# The expression of established cognitive brain states stabilizes with working memory development

David Florentino Montez[1,2]*, Finnegan J Calabro[1,2,3], Beatriz Luna[1,2]

[1]Department of Psychiatry, University of Pittsburgh, Pittsburgh, United States; [2]Center for the Neural Basis of Cognition, Pittsburgh, United States; [3]Department of Bioengineering, University of Pittsburgh, Pittsburgh, United States

**Abstract** We present results from a longitudinal study conducted over 10 years in a sample of 126 8–33 year olds demonstrating that adolescent development of working memory is supported by decreased variability in the amplitude of expression of whole brain states of task-related activity. fMRI analyses reveal that putative gain signals affecting maintenance and retrieval aspects of working memory processing stabilize during adolescence, while those affecting sensorimotor processes do not. We show that trial-to-trial variability in the reaction time and accuracy of eye-movements during a memory guided saccade task are related to fluctuations in the amplitude of expression of task-related brain states, or brain state variability, and also provide evidence that individual developmental trajectories of reaction time variability are related to individual trajectories of brain state variability. These observations demonstrate that the stabilization of widespread gain signals affecting already available cognitive processes underlies the maturation of cognition during adolescence.

DOI: https://doi.org/10.7554/eLife.25606.001

*For correspondence:
dfm11@pitt.edu

Competing interests: The authors declare that no competing interests exist.

## Introduction

Working memory (WM), the ability to retain information online in order to guide goal directed behavior, is evident in rudimentary form as early as infancy (*Diamond and Goldman-Rakic, 1989*; *Gilmore and Johnson, 1995*), indicating that core WM processes are available throughout development. However, protracted improvements in performance, typically measured as the percentage of correctly performed trials or as changes in mean behavioral response metrics, such as reaction time and accuracy, demonstrate that WM continues to develop across adolescence and into early adulthood (*Gathercole et al., 2004*; *Luna et al., 2004*; *Alloway et al., 2006*; *Crone et al., 2006*; *Thomason et al., 2009*; *Luna et al., 2015*). Developmental decreases in behavioral variability during cognitive tasks are also evident through adolescence (*McIntosh et al., 2008*; *Klein et al., 2011*; *Tamnes et al., 2012*), however this has not been directly examined in the context of WM.

Behavioral variability, or intra-individual variability, is a sensitive barometer of cognitive function; excessive variability frequently attends disorders such as schizophrenia (*Kaiser et al., 2008*); ADHD (*Munoz et al., 2003*); and age-related cognitive decline (*MacDonald et al., 2006*). The association between behavioral variability and cognitive performance suggests that adolescent stabilization of behavior reflects the continuing alteration of fundamental aspects of brain processing that support the transition to adult-like levels of performance. Mechanistically accounting for the stabilization of behavior is critical to our understanding of adolescent neural development. Thus, in the present study, we examine the neural processes that underlie behavioral *variability*.

Behavioral variability has been found to be associated with fluctuations in neural (*Newsome et al., 1989*; *Cohen et al., 2009*; *Nienborg and Cumming, 2009*) or blood-oxygen-level

**eLife digest** Adolescence is a period of change: physically, socially and intellectually. During the teenage years, the brain undergoes changes in structure and connectivity that lead to improvements in areas such as self-control, social skills and cognition. Adolescence is also a time during which cognitive skills, such as problem solving and memory, become more stable. Whereas a child will perform a task markedly better on some days or trials than others, adolescents become increasingly consistent.

But why does cognitive performance fluctuate at all? Studies in monkeys suggest that momentary fluctuations in processes like attention and alertness are linked to changes in the level of activity within brain regions that are active during a task. Areas of the brain that are relatively active tend to become even more active, whereas those that are relatively inactive reduce their activity even further. These changes lead to variable accuracy and reaction times. Montez et al. hypothesized that as adolescents become better at controlling processes such as attention and alertness, they show fewer and/or smaller fluctuations in brain-wide activity during a task. This in turn leads to more stable performance.

To test this idea, Montez et al. asked healthy volunteers aged 8 years and above to perform a memory task while lying inside a brain scanner. Over the next 10 years, the volunteers returned about once a year to perform the task again, thereby revealing how their brain activity changed as they grew older. Over the course of adolescence, the volunteers performed the task increasingly accurately and consistently. As predicted, their overall level of brain activity during the task also became less variable over the same period.

These findings challenge the current view of adolescent development, which assumes that teenagers acquire new cognitive skills with age. The results of Montez et al. suggest instead that improvements in cognitive performance reflect teenagers' increasing ability to stably engage skills that they have possessed since childhood. This difference has implications for education, healthcare, parenting, and even for the juvenile justice system.

DOI: https://doi.org/10.7554/eLife.25606.002

dependent (BOLD) signals occurring within individual brain areas (*Wagner et al., 1998*; *Ress and Heeger, 2003*; *Yarkoni et al., 2009*), and networks of brain areas (*Rosenberg et al., 2016*). Several lines of research suggest that these brain/behavior relationships are driven, at least in part, by trial-to-trial variability in gain modulating signals (*Fox et al., 2006*; *Eldar et al., 2013*; *Rabinowitz et al., 2015*) that enhance the activity of individual neurons or brain areas experiencing net excitation and further suppress the activity of neurons or areas experiencing net inhibition (*Servan-Schreiber et al., 1990*). Recent electrophysiological evidence indicates that some gain modulating signals are shared across cortical areas and that moment-to-moment variation in such distributed gain signaling may account for a significant portion of neural and behavioral variability (*Rabinowitz et al., 2015*). Additionally, the structure of neural covariance within populations of simultaneously recorded sensory neurons is best explained as the result of multiple ongoing gain modulating signals (*Rabinowitz et al., 2015*), raising the possibility that different sources of gain variability affect neural activity in a functionally targeted way.

Widespread or global gain modulating processes would not, by definition, change the spatial distribution or 'pattern' of task evoked neural activity. Rather, they would influence the *amplitude* with which they are expressed, manifesting as trial-to-trial variability in the amplitude of expression of whole-brain patterns of activity that support task-relevant processes. We refer to this hypothesized phenomenon here as 'brain state' variability. With this operationalized definition of gain modulation, once an average pattern of task-evoked activity is known, the occurrence of fluctuations in global gain signals may be determined by measuring trial-to-trial differences in the amplitude of expression of the average whole-brain task state.

In the present developmental functional magnetic resonance imaging (fMRI) experiment, we exploit this anticipated characteristic of global gain modulation to study the relationship between trial-to-trial variability in behavioral responses during a memory-guided saccade (MGS) task and trial-to-trial variability in widespread gain signals (i.e., brain state variability) that occur near the time of

the behavioral response. We explore the possibility that the reduction of behavioral variability observed during development is the result of stabilizing widespread gain signals.

We performed fMRI on an accelerated longitudinal cohort of 126 subjects between the ages of 8 and 33 years (*Figure 1a*) as they performed a variant of the MGS task (*Hikosaka et al., 1989*) (*Figure 1b*). On each trial, subjects first made a visually guided 'encoding' saccade to the target stimulus, which appeared at one of six locations: ±3, 6, or 9° along the horizontal visual meridian, during an initial visuomotor/encoding (VME) epoch. Children typically have difficulty suppressing orienting saccades to target stimuli (*Luna et al., 2004*); by allowing subjects to make the initial encoding saccade, we removed a possible age-related source of behavioral differences that would be related to response inhibition rather than WM performance. After making a saccade to the target, subjects returned their gaze to a central fixation point, marking the onset of the maintenance epoch. The working memory retrieval epoch began when fixation was extinguished and subjects generated a saccade to the remembered location. We varied the duration of the initial target presentation (either 1.5 or 3 s) and the duration of the delay (1.5 or 9 s). We measured the subject's gaze location in the scanner with an MR compatible infrared camera and eye-tracking system (Model R-LRO6, Applied Science Laboratory, Bedford, MA).

We applied several levels of analysis: First we characterized developmental changes in mean behavioral performance and behavioral variability and determined the extent to which measures of behavioral variability constitute a distinct metric of developmental status beyond that provided by measures of mean behavioral performance. Second, we identified canonical whole brain patterns of activity (brain states) corresponding to visuomotor, WM maintenance, and retrieval processes and

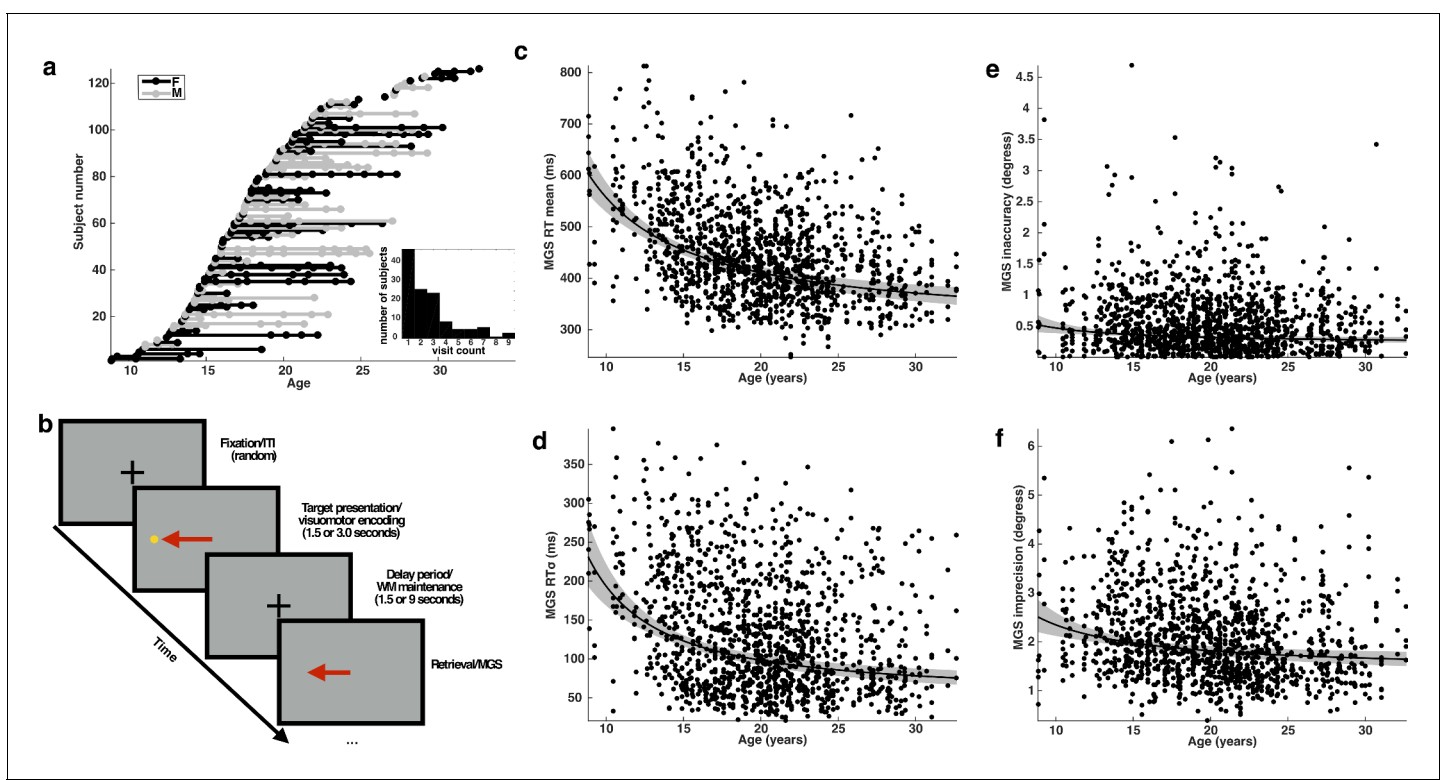

**Figure 1.** Subjects, task, and behavioral performance. (**a**) The distribution of subjects included in our analyses. Each entry on the y-axis represents a unique subject. Each visit is indicated by a dot whose x-coordinate corresponds to the age of the subject at that visit. A solid line connects repeated sessions. Female subjects are rendered in black and males in gray. Inset histogram depicts the distribution of the number of single and return visits across subjects. (**b**) A schematic depiction of the variant of the memory-guided saccade task employed in our study. Developmental changes in behavioral performance for each task condition across age: (**c**) mean reaction time; (**d**) reaction time variability (standard deviation); (**e**) inaccuracy (average absolute value of SE); and (**f**) imprecision of memory-guided saccades (standard deviation of SE). Each black curve depicts the best fitting group-level age$^{-1}$ trajectory. The gray envelope represents the 95% confidence bounds.

DOI: https://doi.org/10.7554/eLife.25606.003

determined whether these states were similarly expressed across development. Third, we measured the relationship between trial-to-trial fluctuations in the amplitude of expression of the task-related brain states and single trial behavioral performance. Fourth, we examined developmental trajectories of brain state variability and its relationship with individual developmental trajectories of behavioral variability.

## Results

### Behavioral performance improves and stabilizes through development

For each trial, we assessed two measures of performance: reaction time (RT), the interval between the extinction of the fixation stimulus at the end of the delay interval and the initiation of the MGS, as well as saccadic error (SE), the signed visual angle separating the horizontal location of the target and the end point of the MGS. For the four task conditions during a session, we computed average RT and the standard deviation of RT. As additional measures of mean behavioral performance and behavioral variability, we defined saccade inaccuracy as the absolute value of the average SE for a given target and saccade imprecision as the standard deviation of SE for each target.

Using a linear mixed-effects model to account for the longitudinal nature of the data, we found, consistent with prior findings (*Luna et al., 2004*), that average RT and inaccuracy decreases during development following an age$^{-1}$ trajectory (*Figure 1c,e*). We estimated how much each behavioral measure changed at the group level between the ages of 8 and 33 and observed a reduction in average RT amounting to approximately 239 ms, a 39% decline (t(1346)=9.30; p=6.03e-20 Inaccuracy decreased by approximately 0.25°, or 47% (t(1346)=3.11; p=0.0019).

Importantly, both measures of behavioral variability also decrease with development following an age$^{-1}$ trajectory (*Figure 1d,f*). The developmental change in the standard deviation of RT amounts to approximately 156 ms, a 67% reduction (t(1346)=7.84; p=9.4e-15). The imprecision of the MGS decreases with age, resulting in an estimated reduction of 0.87° of visual angle, a change, which amounts to roughly 35% (t(1346)=4.03; p=5.83e-5).

In order to determine whether behavioral variability provides additional information beyond mean behavioral measures about the developmental status of a subject, we compared the performance of linear models that predicted subject age from either mean RT or inaccuracy (null models) to matched linear models containing the corresponding behavioral variability factor (full models). We found that a null model predicting age from only mean RT was significantly improved by including the standard deviation of RT (null model: DF=4 AIC=−2212.7, Log-Likelihood=1110.4; full model: DF=5 AIC=−2220.7, Log-Likelihood=1115.6; p=0.002)(*Burnham and Anderson, 2004*). Including imprecision as a term improved model performance, but the difference did not achieve statistical significance (null model: DF=4, AIC=−2108.4, Log-Likelihood=1058.2; full model: DF=5 AIC=−2110.1, Log-Likelihood=1060.0, p=0.052). Thus, of the two measures of behavioral variability, reaction time variability, but not imprecision, appears to measurably reflect a unique aspect of cognitive development.

### Transforming voxelwise time courses of task-evoked BOLD signal into time courses of brain state expression

Performance of the MGS task requires coordinated activation of many brain regions involved in visuomotor, WM maintenance, and retrieval processes. Each of these processes is associated with a distinct whole-brain pattern of activity, or 'brain state', which is expressed at the appropriate times during a trial. Developmental changes in average global gain would result in changes in the average amplitude of expression of task-related brain states across trials; subjects with greater mean global gain would express the task-related brain states to a greater extent, while subjects with reduced global gain would express the brain states with reduced amplitude. We therefore sought to estimate the canonical brain state patterns associated with the visuomotor, maintenance, and retrieval processes involved in the MGS task and to determine whether adolescent development is associated with changes in the mean amplitude of their expression.

Using a simple dimensionality reduction technique based on linear regression (see Materials and methods), we constructed whole brain patterns of activity, that is, brain states, associated with visuo-motor/encoding (VME), WM maintenance, and retrieval processes. These task-related brain state

patterns were extracted from idealized time courses of BOLD activity (see Materials and methods) observed during the long delay trials (*Figure 2a*).

To represent VME processes we extracted the average patterns of BOLD signal occurring 6 s after the initial 'encoding' saccade. To represent retrieval processes, we again extracted patterns of activity present 6 s after the MGS. This delay allowed the BOLD signals associated with these processes to reach their peaks (*Handwerker et al., 2004*). The pattern of activity associated with working memory maintenance was extracted from the TR immediately prior to the execution of the MGS, allowing us the purest estimate of delay period activation (furthest from the preceding visually-guided saccade, without intruding into the subsequent MGS). We orthogonalized each of the brain state patterns to ensure that they captured unique aspects of task activity by regressing the VME-related patterns from maintenance-related patterns, and regressing both VME- and maintenance-related patterns from the retrieval related patterns. This process removed remaining components of VME-related activity from the maintenance activity and importantly, allowed us to remove the pattern of activity associated with visuomotor responses from the retrieval-related pattern occurring during the MGS. Implicit in this procedure is the assumption that VME, maintenance, and retrieval processes are associated with distinct and consistent patterns of whole-brain BOLD activation that are expressed with the time course of a hemodynamic response.

Some neuronal gain modulators, particularly those acting through cholinergic pathways, alter gain with hemispheric specificity, similar to the effects of directed spatial attention (*Sarter and Bruno, 1997*; *Furey et al., 2000*; *Bentley et al., 2004*). To account for such potential hemispheric differences in gain signaling, we decomposed each brain state into two component patterns: a target hemifield specific 'spatial' component and a target hemifield non-specific, 'mean' component. Mean brain state components correspond to the average whole-brain patterns of activity associated with VME, maintenance, or retrieval, regardless of which visual hemifield the target was presented in (*Figure 2b* upper panels). Spatial components were constructed by computing the difference —right minus left— between brain state patterns determined separately for right and left side targets; they reflect the differences in activity during each task epoch resulting from the changes in the target's visual hemifield (*Figure 2b* lower panel). The whole brain patterns of activity observed during different epochs of the task can therefore be approximated with linear combinations of the mean and spatial components of the brain state patterns. For instance, maintenance related activity observed during trials in which the target appears in the right visual hemifield, can be approximated by *adding* the mean and spatial components of the maintenance brain state patterns, while maintenance activity observed during trials in which the target appears in the left visual hemifield can be approximated by *subtracting* the spatial component of the maintenance brain state from the mean component.

Taken together, the brain states characterize the patterns of engagement of canonical regions underlying the VME epoch (e.g., frontal eye fields), maintenance (e.g., prefrontal and frontal eye fields) and the non-visuomotor aspects of retrieval and response (e.g., preSMA) (*Anderson et al., 1994*; *Fried et al., 1991*).

We verified that the resulting brain state components captured the relevant whole-brain patterns of BOLD signal associated with specific task epochs, by projecting each whole-brain volume of a subject's average trial time course, onto the complete set of brain state components using linear regression (see Materials and methods). We performed this operation separately for each task condition. By temporally ordering the regression weights associated with each brain state component from each TR of the average trial time course, we converted the whole-brain time courses of task activity into time courses of task-related brain state expression (*Figure 3*). This procedure is conceptually similar to principle component/independent component analyses in which whole-brain voxel-wise time series are converted into time courses of expression of whole-brain patterns. Here, however, the individual components, that is, brain states, have known behavioral and cognitive processes with which they have been empirically associated. These brain state patterns, although derived from only the long delay trials, also served as an effective basis for describing the whole-brain patterns of BOLD signal evoked during the short delay trials as well (*Figure 3*; lower panel).

## Mean global gain does not change through development

As expected, given our approach for constructing the brain states patterns, the time courses of brain state expression for all subjects exhibited similar temporal characteristics. However, it remained

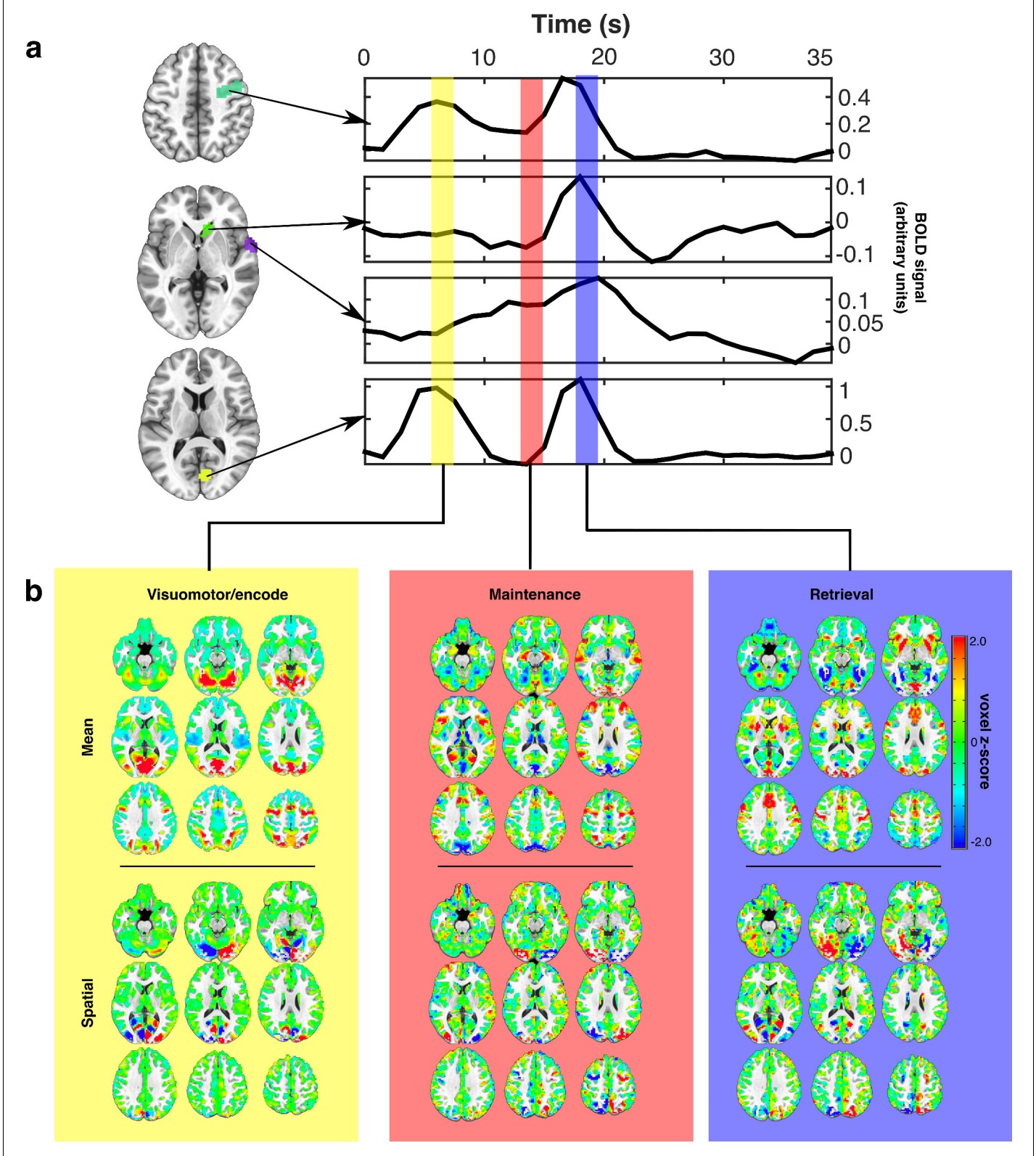

**Figure 2.** Estimating canonical task-related brain states. (a) Representative idealized time series from long presentation, long delay trials where the target appeared in the left visual hemifield. Example ROIs, selected for illustration, are depicted as colored patches on the anatomical underlay and include (from top to bottom) right FEF, caudate head, superior temporal gyrus, and Broadmann area 17. Each colored vertical bar overlay represents a time point that was used for constructing the set of canonical task-related brain state patterns representing VME (yellow), maintenance (red), and

*Figure 2 continued on next page*

*Figure 2 continued*

retrieval (blue) processes. (**b**) The set of task-related brain state patterns derived from the time points indicated above and sequentially orthogonalized. Each brain state consists of two components: mean (top row) and spatial (bottom row). Different linear combinations of mean and spatial components model differences in activity across the brain related to whether the target is on the left or right. Whole brain patterns of task evoked BOLD activity can be converted into time courses of brain state expression by projecting each whole brain volume onto the complete set of brain state patterns using linear regression and organizing the regression weights into a time series.

DOI: https://doi.org/10.7554/eLife.25606.004

possible that there might be age-related differences in mean global gain, which would affect the average *amplitude* of brain state expression. To determine whether such differences were present, we performed omnibus F-tests on the brain state time courses for each task condition to assess the null hypotheses that all of the coefficients for *Age*TR* (*Age*TR*TargetHemifield* in the case of the spatial brain state components) were equal to zero. We observed minimal age-related differences in the average time courses of brain state expression. These differences were observed for the spatial, but not mean, component of the VME states across all trial types (all p<0.001). We did not detect any significant age-related differences in the expression of either mean or spatial components of the maintenance state (all p>0.14). Results for age-related differences in the time courses of expression for the retrieval states were mixed: We observed no omnibus age-related differences within either of the long delay conditions, but within the long presentation short delay trials we observed a small age-related difference in the expression of the spatial retrieval state (F(42,14784)=1.6; p=0.012). Post-hoc examination of the individual *Age*TR* and *Age*TR*TargetHemifield* coefficients at each time point in the trial revealed that this effect was driven by slightly greater expression of the state by adults, across right and left side targets, during the 5th, 6th, and 8th TRs. However, in our post-hoc analysis no single time point reached significance (minimum(p)=0.055). We also observed that the mean retrieval state was differentially expressed across age within the short presentation short delay trials (F(40,14112)=2.0; p<0.001). Post-hoc analyses revealed that this effect was driven by a slightly greater expression of the mean retrieval state by adults during this condition during the 9th-13th TRs, well after the occurrence of peak expression for this state.

From visual inspection it is clear that adults exhibit a slightly prolonged expression of the spatial component of the VME brain state during the different trials (seen most prominently in *Figure 3*; lower panel). We also wanted to know whether the peak amplitude of spatial VME expression differed with age. From each session we examined the amplitude of peak expression of the spatial VME state for each trial type. Because the sign of expression of the spatial VME state varies depending on target hemifield, we extracted the maximum value of positive expression for right side trials, and we extracted the minimum value of expression for left side trials. If adults expressed the spatial VME state to a greater extent than children and adolescents due to greater average gain, this would result in greater positive expression for right side trials and reduced (more negative) peak expression during left side trials. We therefore examined the *Age*Target* interaction term, which we found did not reach significance (t(2696)=1.59; p=0.111).

Combined, these results demonstrate that the set of brain state patterns provide a simplified low dimensional basis for describing BOLD signal changes evoked by the memory-guided saccade task. Importantly, age-related differences in the expression of the brain state patterns during task performance were minimal, and only the spatial component of the VME state exhibited consistent age-related differences in expression across trial types. Even here, however, the age-related differences were not ones of amplitude, but of duration, suggesting that mean global gain does not change during adolescent development.

## Trial-to-trial reaction time and accuracy are associated with fluctuations in global gain that affect the amplitude of expression of brain states

We hypothesized that behavioral performance is affected by fluctuations in global gain signals when they occur around the time of a behavioral response. The signature of variability in global gain signaling would be 'brain state variability', or momentary fluctuations in the amplitude of expression of whole brain states of activity associated with ongoing neural processes.

However, due to the delayed and prolonged nature of the BOLD response, the activity measured near the time of the MGS consists of multiple superposed states of activity associated with

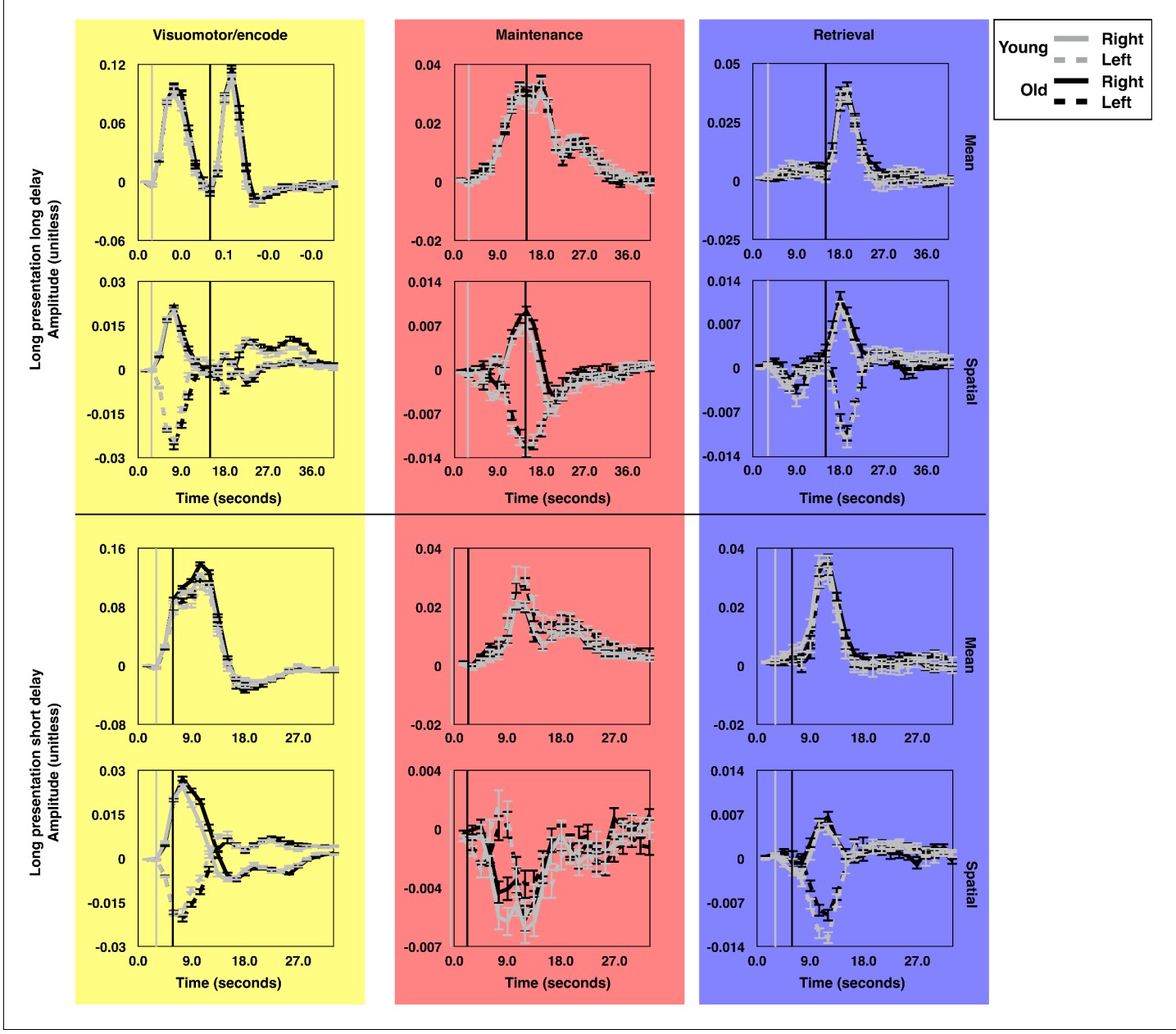

**Figure 3.** The average time course of brain state expression by age group. The average time courses of expression for each brain state component during long presentation, long delay trials (top subset) and short presentation short delay trials (bottom subset). Each panel depicts the average time courses for the oldest (19.9 < Age ≤ 32.6; gray lines) and youngest (8.83 ≤ Age ≤ 19.9; black lines) halves of the subject pool. The time courses for trials during which targets were presented in the left (dashed) and right (solid) visual hemifields are rendered separately. The gray and black vertical lines indicate the time of visually-guided and memory-guided saccade signals respectively. As in *Figure 2*, yellow, red, and blue background colors correspond to VME-, maintenance- and retrieval-related brain state components. Error bars depict one standard error of the mean. These figures depict the absence of significant age related differences in the mean amplitude of expression of the canonical task-related brain states.

DOI: https://doi.org/10.7554/eLife.25606.005

visuomotor, WM maintenance, and retrieval processes. Each of these processes may be affected differently by global gain variability and variability affecting each process might differentially affect behavioral performance. We therefore examined the trial-wise relationship between behavioral performance and the expression of the mean and spatial components of the VME, maintenance and retrieval brain state patterns in an interval of time centered on the occurrence of each MGS.

After removing the mean trial responses from each voxel, we looked for remaining patterns of activity across the brain that matched each of the canonical brain states. To accomplish this, we projected the whole-brain pattern of BOLD signal residual values at each TR onto the set of canonical brain state patterns using linear regression. This approach converted the whole-brain residual time series into a time series of task-related brain state fluctuations, and allowed us to determine the extent to which each brain state component was over- or under-expressed at a particular point in time.

We divided the resulting fluctuation time series for each brain state component into snippets, intervals of time centered on each MGS and extending ±15 TRs (each TR=1.5 s) before and after (*Figure 4a and b*). Each snippet was associated with the RT and SE of a particular trial. After aligning snippets from each trial to the TR in which the MGS occurred (TR 0), we used regression models to measure the relationship between both trial-to-trial differences in RT (z-scored) and SE (z-scored and rectified) and trial-to-trial differences in the expression of each brain state component at different times relative to the MGS. In order to measure the contribution of brain state variability to behavioral variability, we first estimated the fraction of trial-to-trial behavioral variance accounted for by a null model that included terms for non-neural trial-to-trial factors (e.g. target eccentricity) and performance factors (e.g. RT when testing SE and vice versa). Then we measured the additional fraction of behavioral variability that could be explained by including the measures of brain state expression from each TR in the snippets. The difference in explained variance between the full and null models indicates the fraction of trial-to-trial behavioral variability uniquely associated with brain state

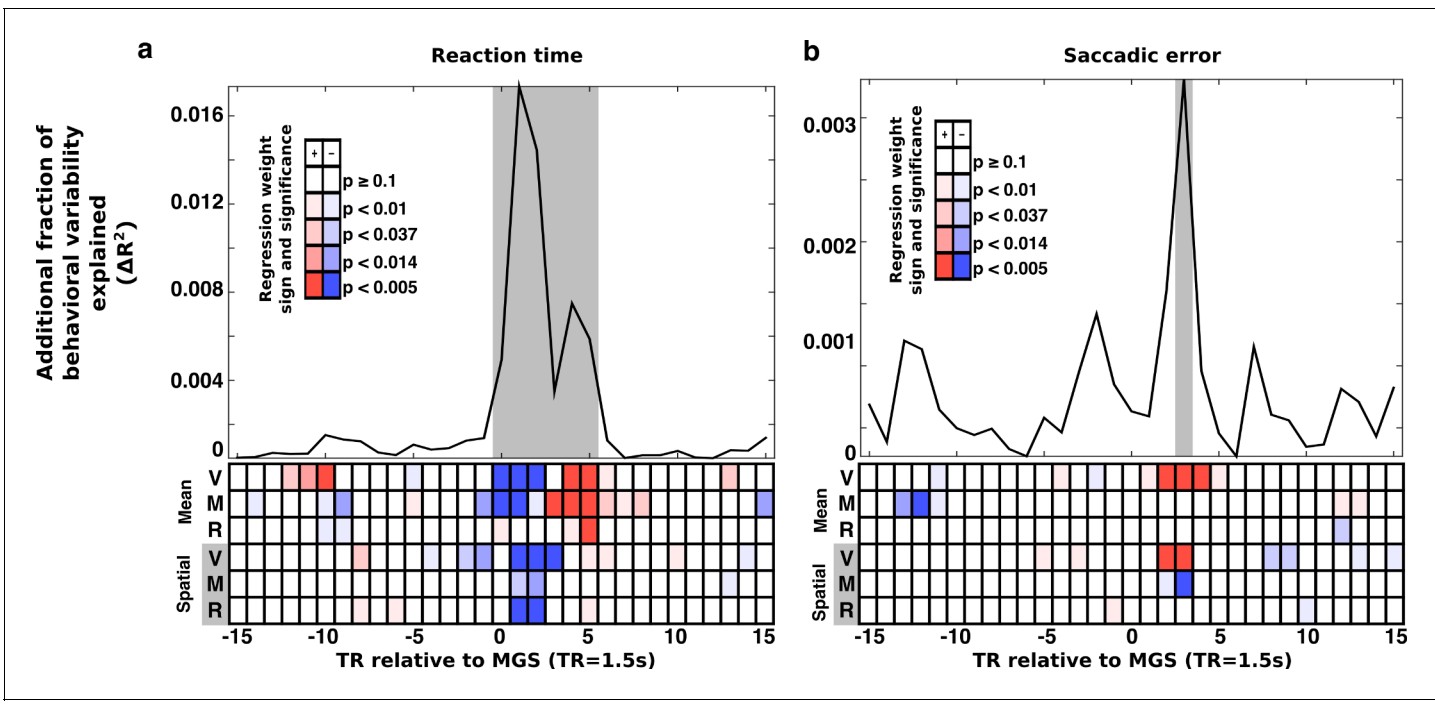

**Figure 4.** Trial-wise relationship between brain state expression and behavior. Trial-to-trial brain state variability is associated with (**a**) reaction time and (**b**) saccade error. For all trials, snippets of the time course of brain state variability, centered on the TR containing the MGS are extracted and aligned (±15 TRs). At each TR relative to the saccade a regression model estimated the relationship between trial-to-trial behavioral performance and variability in the amplitude of expression of the canonical brain state components. The top panels depict the additional fraction of behavioral variability (compared to a null model) accounted for by including brain state measurements from each TR relative to each correct MGS. Highlighted gray intervals represent TRs where the full models provided a better fit than the null model (simulated likelihood ratio test with 5000 iterations; p<0.001). Each cell in the lower panels of (**a**) and (**b**) represent the p-value (darker colors are more significant) and the sign of the regression coefficient (blue, negative; red, positive) for each brain state component. Positive coefficients indicate that greater expression of a brain state component at a given time is associated with an increase in RT or SE. A negative coefficient indicates that greater expression of a brain state component is associated with a reduction in RT or SE. The top three rows of cells represent the mean brain state components (V: VME; M: maintenance; R: retrieval). The bottom three rows represent the significance of interactions between the spatial brain state component and the target hemifield.
DOI: https://doi.org/10.7554/eLife.25606.006

variability occurring at a particular time relative to the execution of a MGS (*Figure 5a and b* [top panels]).

As hypothesized, the relationship between brain state expression and both RT and SE peaked around the time that the MGS was executed. For trial-wise RT, brain state/behavior associations were significant beginning with the TR when the MGS was executed, peaking 1 TR after the saccade, and lasting for a total of 6 TRs. The relationship between trial-wise SE and brain state expression was similar, but much less prominent and evident during only the third TR following the MGS.

Trials with faster RTs were associated with greater early expression of the mean VME and maintenance brain states (TRs 0–2), and reduced later expression of all mean states (TRs 3–5). Greater SE was associated with greater expression of the mean VME state (TR 3).

RT and SE also covaried with fluctuations in the amplitude of expression of the spatial components of the brain state patterns (bottom three rows of *Figure 4a and b*). The fastest RTs occurred when whole-brain patterns of task-related activity were biased in the direction of the target (VME: TRs 1–3; maintenance and retrieval: TRs 1–2). That is, the fastest right side trials were those in which brain states were expressed most 'rightwardly', and the fastest left side trials were those in which the spatial brain states exhibited the greatest 'leftward' expression. Interestingly, greater target hemifield appropriate expression of the spatial VME state (TRs 2–3) was associated with increased SE, indicating that increasing the amplitude of its expression does not improve all aspects of task performance. Increased hemifield appropriate expression of the spatial component of the maintenance states (TR 3), in contrast, was associated with reduced SE.

That greater expression of VME brain states was associated with faster RT and greater SE, prompted us to examine the behavioral data for signs of a speed-accuracy trade-off. We found a

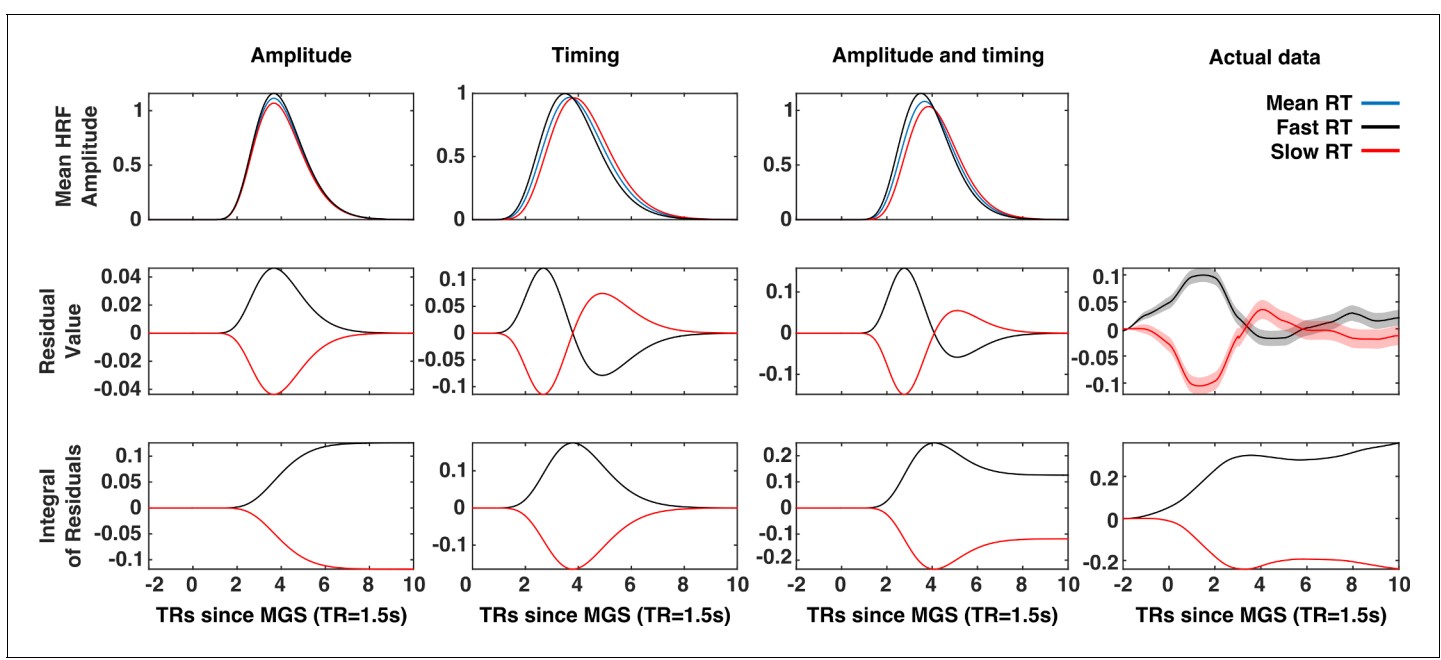

**Figure 5.** A comparison of timing and amplitude effects on brain state expression. The x-axis of each panel represents time (in 1.5 s TRs) relative to the execution of the memory-guided saccade. An amplitude relationship (First column), a timing relationship (second column), and a combined amplitude and timing relationship (third column), are compared to actual data (fourth column). The first row depicts the patterns of trial-to-trial BOLD signal variability for each mechanism. Trials are divided into fast (black) and slow reaction time (red) sets defined by a median split. The average BOLD signal across all trials depicted in blue. The average residuals time series (second row) for fast and slow trials exhibit distinct patterns for each possible mechanism. The time integral of the average residuals makes this difference explicit: If a trial-to-trial relationship between the VME brain state and reaction time simply reflected trial-to-trial variability in the latency of the eye-movement and eye-movement evoked visual activity, then the integrals of the residual time series for fast and slow reaction time trials should both converge to zero (second column). However, the absence of this convergence in the mean VME brain state data (fourth column) is consistent with either an amplitude-based relationship or a mix of amplitude and latency. Where present, red and black envelopes represent ±1 SE of the mean.
DOI: https://doi.org/10.7554/eLife.25606.007

significant quadratic relationship between z-scored RT and SE at the trial level indicating that, within a session, excessively fast and slow responses were associated with reduced accuracy (t(16754) =4.64; p=3.52e-6). Collectively, these results demonstrate that, trial-to-trial, both reaction time and accuracy of subjects' responses covary with the amplitude of expression of whole-brain patterns of task related activity.

## Brain state variability reflects fluctuations in amplitude, not just timing, of brain state expression

Greater early expression and reduced later expression of the mean brain states for fast RT trials (represented by the transition from blue to red in some rows of the lower panel of *Figure 4a*), could possibly be explained by trivial correlation between the timing of a saccade and the latency of the expression of the brain state. That is, on trials with longer RTs, the BOLD activity would be shifted later, causing the brain state to appear under-expressed early and over-expressed late relative to the average time course. To determine whether this relationship could account for the observed correlation with behavior, we performed a set of simulations to compare the temporal patterns of BOLD signal residuals for fast and slow RT trials that would result from timing-, amplitude-, and timing and amplitude-based relationships (see Materials and methods).

As depicted by *Figure 5*, the contributions of timing and amplitude variability to brain state variability are distinguished by their effect on the temporal structure of the residual brain state time series. A purely timing based explanation of the mean brain state/RT relationship predicts that the integral of the mean brain state residual time series, for both fast and slow reaction time trials should converge to zero (*Figure 5*, column 1). A relationship between reaction time and brain state expression mediated by fluctuations in the amplitude of expression of the brain state patterns predicts that the same time integrals converge to non-zero values of opposite sign (*Figure 5*, column 2). A combination of timing- and amplitude-based relationships predicts an initial bifurcation of the time integrals of the fast and slow reaction time residuals that then a partial re-convergence (*Figure 5*, column 3). We found that the trial-wise relationship between RT and the expression of the mean VME brain state —the state associated with eye-movements— was inconsistent with both a purely amplitude-based mechanism, and purely timing-based mechanism and instead is likely to reflect a mixture of both the trivial time shifted BOLD response, and a true relationship between gain fluctuation and performance (*Figure 5*, column 4).

## Brain state variability decreases with development

If the stabilization of behavior during adolescence is related to a reduction in brain state variability, then the proportion of BOLD signal variability associated with brain state variability will decrease with age. We examined the subset of TRs (0–5; the highlighted region *Figure 4a*) around each correct MGS that showed a significant relationship with trial-to-trial behavioral performance and computed $SS_{brain}$, the sum of whole-brain squared error associated with all brain state patterns across each TR as well as $SS_{error}$, the sum of the remaining squared error. For each session, we computed the ratio $SS_{brain}/(SS_{brain} + SS_{error})$, producing values we refer to here as total brain state variability, which corresponds to the fraction of residual whole-brain BOLD signal variability associated with brain state variability. We found that total brain state variability decreases with age (*Figure 6a*) after controlling for mean frame-wise displacement (FD)(*Power et al., 2012*; *Siegel et al., 2014*) and the number of correctly performed trials (t(333)=-3.35; p=9.0e-4).

## Brain state variability related to maintenance and retrieval processes decreases with development

The measure of total brain state variability, $SS_{brain}$, can be simply decomposed into a sum of contributions from the mean and spatial components of VME, maintenance, and retrieval brain state patterns; $SS_{brain} = SS_{VME} + SS_{SMaint} + SS_{Retrieval}$. We computed the ratio of each brain state component sum of squared error to $(SS_{brain} + SS_{error})$, and found that the stability of each of the three sets of brain state components exhibit distinct developmental trajectories (*Figure 6b*). The proportion of VME-related brain state variability did not significantly decrease with age (t(333)=-1.47; p=0.14). However, maintenance- and retrieval-related brain state variability both show significant age-related decreases (t(333)=-3.8; p=1.8e-4 and t(333)=-5.53; p=6.27e-8 respectively). Pairwise comparisons of

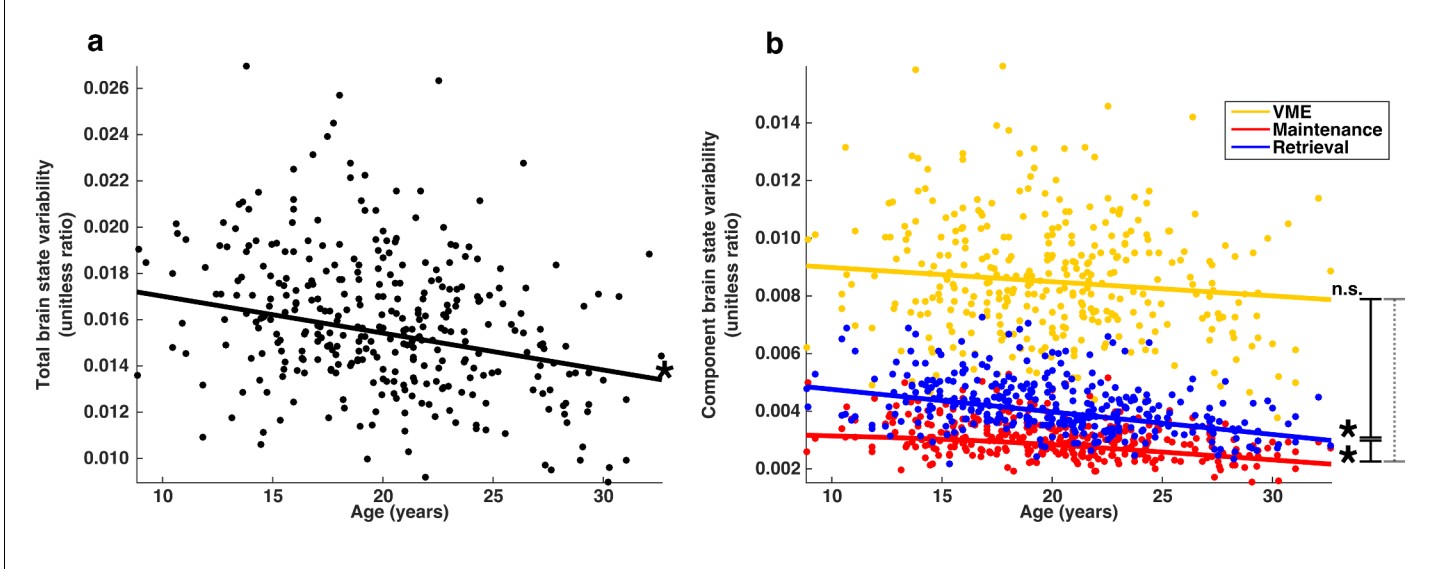

**Figure 6.** Developmental trajectories of brain state variability. (**a**) Trajectory of total brain state variability, the ratio $SS_{brain}/(SS_{brain} + SS_{error})$ across age. A single point represents each session. (**b**) Depicts the developmental trajectories of the components of brain state variability: VME (yellow), maintenance (red) and retrieval (blue). Total brain state variability equal to the sum of the variability of each component. Single asterisks indicate significant individual slopes of brain state component variability; vertical lines connecting endpoints represent pairwise comparisons of slopes. Solid and dashed lines indicate a significant, and non-significant differences respectively.

DOI: https://doi.org/10.7554/eLife.25606.008

slopes reveal that trajectories of VME and retrieval-related variability can not be significantly distinguished from one another ($t(670)=0.60$; $p=0.54$), while maintenance-related brain state variability decreases more slowly through adolescence compared to both VME and retrieval-related variability ($t(670)= 2.44$; $p=0.014$ and $t(670)=-3.71$; $p=2.2e-4$ respectively).

## Age-related changes in brain state variability are not related to motion

Next, we sought to determine whether a systematic relationship exists between in-scanner motion and measures of brain state variability. We reasoned that if brain state variability was unrelated to movement-related artifacts, then our finding that brain state variability was reduced in older subjects would still hold if we selectively sub-sampled our data so that we compared a group of adults who moved excessively to a group of children who moved relatively little. The biased subsampling routine that we employed is based on a mean-matching technique that has been described in detail elsewhere (*Churchland et al., 2010*). Briefly, we divided our data into two sets, split at the median age of our sample. We selectively drew samples from the two data sets such that, on average the older sample exhibited greater mean FD than the younger group. We found that reversing the relationship between motion and age does not significantly alter our finding that older subjects exhibited less brain state variability than younger subjects (*Figure 7*).

## Brain state variability predicts individual differences in behavioral variability

Our analyses to this point have shown that trial-to-trial differences in behavioral performance are associated with brain state variability, which stabilizes during adolescent development. If the stabilization of behavior during adolescence is the result of reduced brain state variability, then subjects exhibiting the greatest longitudinal reduction in behavioral variability should be those that exhibit the greatest decline in brain state variability. To determine whether this is the case, we leveraged the longitudinal design of our dataset to examine how individual differences in the developmental trajectories of RT variability and saccade imprecision are related to individual developmental trajectories of brain state variability.

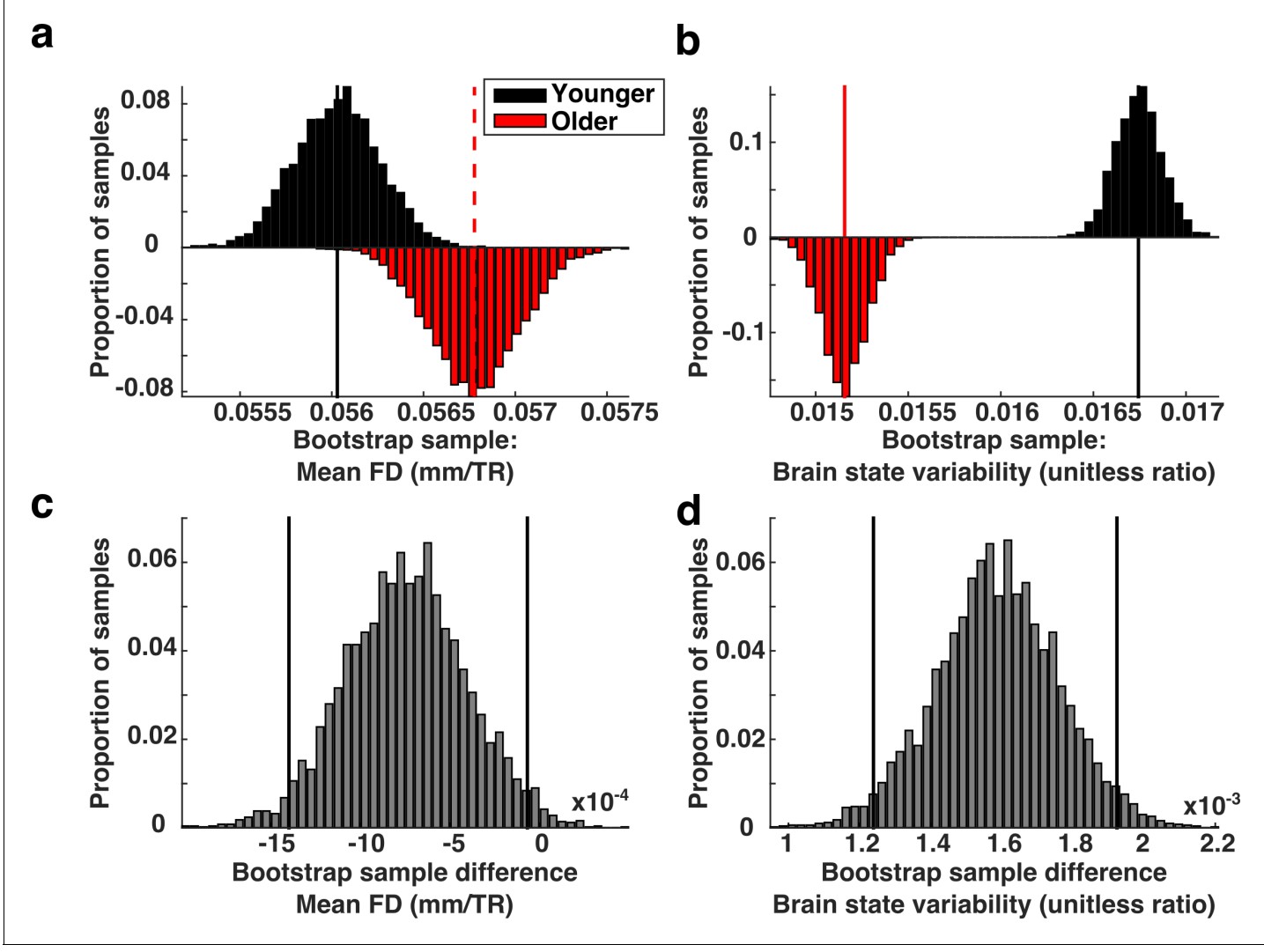

**Figure 7.** Brain state variability in high motion adults compared to low motion children. (a) Biased bootstrap sample distribution of mean FD in which the older subjects (red histogram) moved more than younger subjects (black histogram). (b) The corresponding distributions of brain state variability demonstrate that brain state variability is still greater in younger subjects who moved more than adults. Vertical lines in the top two panels represent the means of the distributions. (c–d) The bootstrap differences for the corresponding distributions above. Vertical black lines indicate 95% confidence intervals.

DOI: https://doi.org/10.7554/eLife.25606.009

We selected a subset of 29 subjects for whom we had at least four complete sessions of data, and estimated individual developmental slopes of total brain state variability. We modeled developmental changes in total brain state variability as a linear effect of age after observing superior performance compared to a model in which total brain state variability was fit with a $age^{-1}$ term (simulated likelihood ratio test with 5000 iterations, $age^{-1}$ model: DF=9 AIC=−3116.2, Log-Likelihood=15567.1; linear age model: DF=9 AIC=−3120.2, Log-Likelihood=1569.1; p=0.015);. We also estimated individual regression weights for the developmental trajectories of reaction time variability and imprecision after controlling for task condition and mean reaction time. Here we used an $age^{-1}$ term to model individual trajectories.

Subjects exhibiting the greatest decreases in behavioral variability given their age have the greatest $age^{-1}$ regression coefficients. In contrast, subjects exhibiting the greatest decrease in total brain state variability have the smallest age regression coefficients. Evidence consistent with the hypothesis that the developmental stabilization of behavior is driven by a reduction in brain state variability

would be the existence of a negative correlation between the brain state and behavioral regression coefficients. We observed just such a negative relationship for reaction time variability (r = −0.48; p=0.008) (*Figure 8a*). This result remained significant (p<0.05) when we limited our data set to subjects with 3–5 or more sessions as well. However, the within-subject relationship between brain state variability and saccade imprecision was not significant (r=0.28; p=0.14) (*Figure 8b*).

Given the modest relationship between brain state variability and SE at the single trial level, we considered the possibility that we might be underpowered to detect the longitudinal relationship between total brain state variability and saccadic imprecision in our smaller sample size. We expanded our analyses to investigate whether individual differences in saccade imprecision were related to individual differences in brain state variability using our entire sample and including age$^{-1}$ terms as covariates. Here we did observe a significant positive relationship between total brain state variability and saccade imprecision (t(1344)=3.2; p=0.001) (*Figure 8c*).

## Discussion

Reduced variability is a key component of the behavioral improvements that are observed during adolescent development. We demonstrated an example of this stabilization using a working memory task in which subjects' performance of memory guided saccades improved on average and became less variable with age. To understand the neural basis of developmentally stabilized behavior, we investigated the relationship between variability in the reaction times and accuracies of eye-movements and fluctuations of global gain signals hypothesized to affect the amplitude of expression of whole-brain states of activity underlying distinct task-related processes. We found that while the average amplitude of expression of whole-brain task states was similar across subjects, regardless of their age, trial-to-trial variability in the amplitude of their expression decreased during adolescence and was correlated with trial-to-trial variability in the reaction time and accuracy of memory-guided saccades. Importantly, this brain state variability represented fluctuations in the *amplitude* of brain state expression across trials, not simply variability in the *timing* of their expression or global fluctuations in mean activity (see Materials and methods).

Additionally, variability occurring specifically in the expression of the mean and spatial components of the VME brain states associated with visuomotor processes mirrored the higher-order phenomenon of speed-accuracy trade-off, that is, greater VME expression was associated with faster responses and increased saccadic error. Greater expression of the spatial component of the working

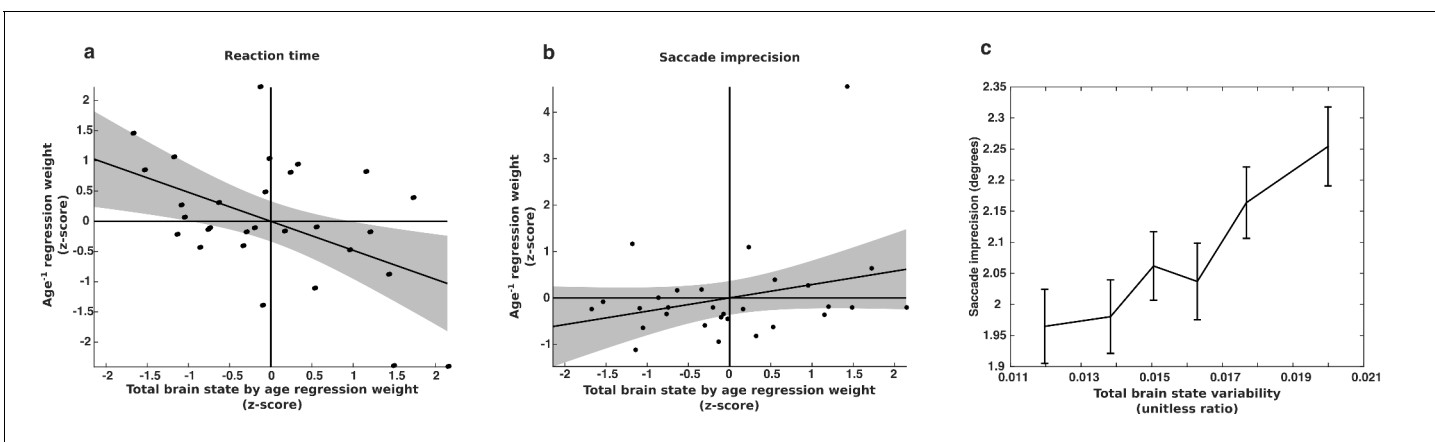

**Figure 8.** Longitudinal relationship between brain state variability and behavioral variability. The relationship between changes in brain state variability and changes in a) reaction time variability and b) memory-guided saccade imprecision for a group subjects (n=29) for whom we had four of more complete sessions of data. The x-axes depict the slopes of age-related change in brain state variability, computed separately for each subject. They y-axes depict the regression weight for the age$^{-1}$ term used to fit each subjects' behavioral data. Thus, a negative relationship indicates that greater reduction in brain state variability is associated with a greater reduction in behavioral variability. (c) The relationship between saccade imprecision and brain state variability using our entire data sample. For presentation purposes, the data was adaptively binned so that each point contains the same amount of data and is centered on the median value for that bin. Error bars represent ±1 standard error of the mean.
DOI: https://doi.org/10.7554/eLife.25606.010

memory maintenance state, in contrast, was associated with faster responses and reduced saccadic error. These findings are broadly consistent with recent theoretical models (*Standage et al., 2011*) and empirical data from non-human primates (*Heitz and Schall, 2012*) suggesting gain modulation plays a role the speed-accuracy trade-off. Appropriately balanced, independent variability in gain signals affecting VME and maintenance brain state expression, may explain the quadratic speed accuracy trade-off that we observed in our data.

We hypothesized that developmental decreases in the variability of global gain signaling would result in more stable expression of task-related brain states. Accordingly, we determined whether the expression of brain states associated with visuomotor/encoding (VME), maintenance, and retrieval processes, exhibited similar or different trajectories of variable expression across development. We found that the variability of the VME states did not decrease with age although they were significant predictors of single trial performance. Our task design did not allow us to dissociate the activity involved strictly in working memory encoding from that involved strictly in the visuomotor response, however, the re-expresssion of the VME states during the memory-guided saccade suggests that they are largely dominated by visuomotor activity. In contrast, working memory maintenance and retrieval processes, whose fluctuations were also related to trial-wise performance, showed significant decreases in the variability of their expression. Perhaps most significantly, we found a relationship between individual longitudinal changes in total brain state variability and changes in reaction time variability as well as a relationship between total brain state variability and memory-guided saccade imprecision after covarying for age. Combined, our findings provide evidence that adolescent developmental changes in behavioral variability are driven by the stabilization of gain signals specifically affecting cognitive processes while gain signals affecting sensorimotor processes continue to vary greatly across all ages.

A complex interplay between top-down control (*Moore and Armstrong, 2003*; *Larkum et al., 2004*) and a mixture of contributions from several interconnected neuro-modulatory systems, each exerting its particular influence on ongoing sensorimotor, and cognitive processes (*Furey et al., 2000*; *Luciana et al., 1998*; *Rokem et al., 2010*; *Chandler et al., 2014*) may underlie these developmental changes in brain state variability. Recent fMRI studies have shown that fluctuations in the activity of midbrain and brain stem nuclei affect resting state connectivity in what appears to be a functionally organized way (*Bär et al., 2016*). Similarly, cholinergic modulation has been shown to amplify the spatially selective effects of perceptual processing and attention in a manner analogous to fluctuations in our spatial brain state components (*Furey et al., 2000*; *Bentley et al., 2004*; *Rokem et al., 2010*). Finally, myelination and synaptic pruning, which continue to progress in critical brain systems (*Simmonds et al., 2014*), occurring at different rates for different brain regions, (*Petanjek et al., 2011*) may also affect neural signal to noise ratios and play a role in the stability of gain signals that contribute to behavioral variability. Differing rates of development in any of these systems could produce distinct developmental trajectories for the components of brain state variability.

The presence of brain state variability also bears upon the interpretation of brain/behavior correlations in general. In studies of single unit and population activity in non-human primates, correlations between the trial-to-trial fluctuations of neuronal activity and behavioral responses, often termed choice-probability (CP) or detect-probability (DP), have been interpreted as signifying a neuron's causal role in the behavior (*Shadlen et al., 1996*). It has been proposed, however, that brain/behavior relationships like CP and DP, might reflect a neuron's covariation with a neuronal gain signals, such as attention, rather than direct causal involvement (*Nienborg and Cumming, 2009*; *Cohen and Kohn, 2011*). Brain state variability is consistent with this hypothesis and expands upon it in two ways 1) That brain state variability is the covariation of many task-related (and presumably behaviorally relevant) brain regions suggests that brain/behavior correlations like CP and DP should be wide-spread throughout task-related brain areas; and 2) Our finding of distinct developmental trajectories of brain state variability affecting different task-related processes suggests that fluctuations in multiple functionally specific global gain signals contribute to observed brain behavior correlations. This interpretation also gains support from recent electrophysiological evidence that multiple independent gain modulating signals are apparent within the activity of populations of neurons in sensory cortex (*Rabinowitz et al., 2015*).

An attractive model that would provide synthesis for our findings and those discussed above is that stable behavioral performance requires similarly stable allocation of top-down control

processes, like attention. Such top-down processes partly exert their influence through multiple widespread gain signals that are functionally targeted, differentially affecting sensorimotor and cognitive processes. In this model, the stabilizing of working memory behavioral performance that we observed during adolescent development is the result of stabilizing those gain signals that affect working memory maintenance and retrieval processes. Our results, in sum, provide compelling evidence that core cognitive functions are online by childhood and what underlies cognitive development through adolescence is a fine-tuning of the ability to stabilize the expression neural activity associated with those cognitive processes.

## Materials and methods

### Subjects
We tested 152 subjects between the ages of 8 and 33. Subjects were initially recruited between the ages of 8 and 30 years and were scanned approximately annually for 1–10 years. Subjects were included based on two criteria: (1) Mean frame-wise displacement (FD) was less than 0.15 mm; and (2) at least 50% of the trials from each of the four trial types had to be measurably correct. Here, incorrect trials are those for which measurements of reaction time and endpoints for both visually- and memory-guided saccades were unavailable due to blink artifacts, noisy data, or transient loss of pupil- or corneal reflection-lock. After applying these exclusion parameters our dataset consisted of 126 subjects (60 female). We applied no further outlier control for our analyses. Participants and/or their legal guardians provided informed consent before participating in this study. Experimental procedures for this study complied with the Code of Ethics of the World Medical Association (1964; Declaration of Helsinki) and the Institutional Review Board at the University of Pittsburgh. Subjects were paid for their participation in the study.

### Eye-movements
Eye-movements were recorded in the scanner with an infrared camera system equipped with long-range optics and sampling at 60 Hz (Model R-LRO6, Applied Science Laboratories, Bedford MA). Subjects' compliance with instructions was assessed and eye-movements were monitored via remote video during task performance. We used a nine-point calibration procedure to estimate the transformation from the eye-tracker's native encoding space to on-screen pixel location. Saccadic events were detected using an in-house suite of automated routines. Individual saccade candidate events were detected from local maxima in the eye-movement velocity trace. Saccade start and end times were determined by searching backward and forward in time in the velocity trace to find the sample where velocity dropped below 1/10th of the peak velocity (*Gitelman, 2002*).

### fMRI data and acquisition parameters
Imaging data were acquired using a Siemens 3-Tesla MAGNETOM Allegra (Erlangen, Germany) system with a standard radio-frequency (RF) head coil at the Brain Imaging Research Center, University of Pittsburgh, Pittsburgh, PA. Structural images were acquired using a sagittal magnetization prepared rapid gradient echo (MPRAGE) T1-weighted pulse sequence with 224 slices with 0.7825 mm slice resolution. Functional images were acquired using a gradient echo echo-planar (EPI) sequence sensitive to blood-oxygen-dependent (BOLD) contrast (T2*) (TR=1.5 s, TE=25 ms, flip angle=70°, voxel size=3.125×3.125×4 mm slice resolution, 229 volumes). Twenty-nine slices per volume were collected with no gap and aligned to the anterior and posterior commissure (AC-PC) plane. Structural and functional fMRI data are available through the Dryad repository (*Montez et al., 2017*).

### Anatomical preprocessing
T1-weighted anatomical images were reconstructed from raw DICOM files and converted to NIFTI format. We estimated the bias field corrections using smoothed and highpass filtered anatomical data analyzed with FSLs *fast* algorithm. After bias field correction we constructed a skull stripped anatomical data set for the subject, which we used to estimate the 12 degree-of-freedom affine transformations that would align the subjects data with the MNI152 anatomical template. Finally, we computed the non-linear transformation that would bring the subject's affine-aligned anatomical

data set into registration with the MNI152 template. We saved final combined linear/non-linear transformation for later use in registering the subjects' functional data to the standard space.

## Functional preprocessing

fMRI data were preprocessed using a combination of AFNI (Analysis of Functional NeuroImages, RRID:SCR_005927) and FSL software (FSL, RRID:SCR_002823). In our pre-processing pipeline, raw data was converted from DICOM format to NIFTI volumes and slice-timing correction was applied using AFNI tools. We performed motion estimation and correction in two phases. First we pre-aligned each frame of a subject's functional data to a volume created by taking the temporal mean of the 4-D functional time series. Then, a second, 'true', average functional volume was computed from the pre-aligned functional data, producing a reference functional volume that was less affected by motion artifacts. We then aligned each frame of the original function time series to this second reference volume using sinc-function interpolation and estimating the time course of translational and rotational motion throughout the run. We used these estimated time series throughout our later analyses of the functional data.

Next, using FSL's brain extraction tool, we stripped the skull and superfluous tissues from the subject's motion corrected mean functional EPI images, afterward aligning the resulting mean EPI volumes to their anatomical MPRAGE volume using a six degree-of-freedom rigid-body transformation estimated using spline interpolation. To align each frame of the motion corrected EPI sequence to the subjects structural image, we applied the translation estimated in the previous step to each frame of the motion corrected functional time series and then removed the skull and extraneous tissues from each frame of the functional time series. Tissue remaining within the mean functional volume after the skull stripping procedure was removed by applying a dilated binary mask to the mean aligned functional volume that removed extreme voxels whose values did not reside in middle 98th percentile. We then removed voxel-wise temporal extrema using AFNI's 3dDespike software.

To align a subjects functional data to a standard MNI152 (Montreal Neurological Institute; MNI) template in a single transformation step, we used FSL *convertwarp*, and *applywarp* functions to combine the estimated motion correction, functional-to-structural, and linear and non-linear subject-to-MNI152 transformations into a single operator, which we applied separately to each frame of the original slice time-corrected functional data.

We performed minimal spatial smoothing on the aligned functional data, using a SUSAN algorithm with a 5 mm FWHM kernel, followed by a conservative high-pass filtering of the voxel-wise time series, which removed or attenuated BOLD signal frequencies below 0.0083 Hz (corresponding to fewer than three cycles per task run). Finally, we rescaled all voxel values by a value defined to be 10,000 divided by the global median.

## Overview of brain state analysis

In our analyses, measuring brain state variability requires determining the spatial structure of canonical whole-brain patterns of BOLD signal associated with distinct task-related processes. The flowchart in *Figure 9* depicts an outline of the processing steps used to transform individual preprocessed fMRI time series into the average time courses of brain state expression and brain state variability as well as the relationship between the major processing steps and certain key analyses.

## Deconvolution

From each session's data we extracted eight voxel-wise average time courses of BOLD activity corresponding to each of the four task conditions when stimulus presentations occurred in wither the left or right visual hemifields. We estimated these time courses with a finite impulse response (FIR) regression model. FIR design matrices were constructed manually and applied to the voxel-wise time series using 3dDeconvolve (AFNI). All trials, including incorrect responses and blinks, for each stimulus type were modeled over an interval consisting of the duration (from initial stimulus presentation to the execution of the memory-guided saccade) plus an additional 22.5 s (15 TRs). The design matrix included nuisance regressors to account for the effects of signal drift, subject motion, and global signal changes as captured by white matter and cerebrospinal fluid (CSF) signals and their derivatives. Signal drift for each run was modeled as a third order Legendre polynomial time series.

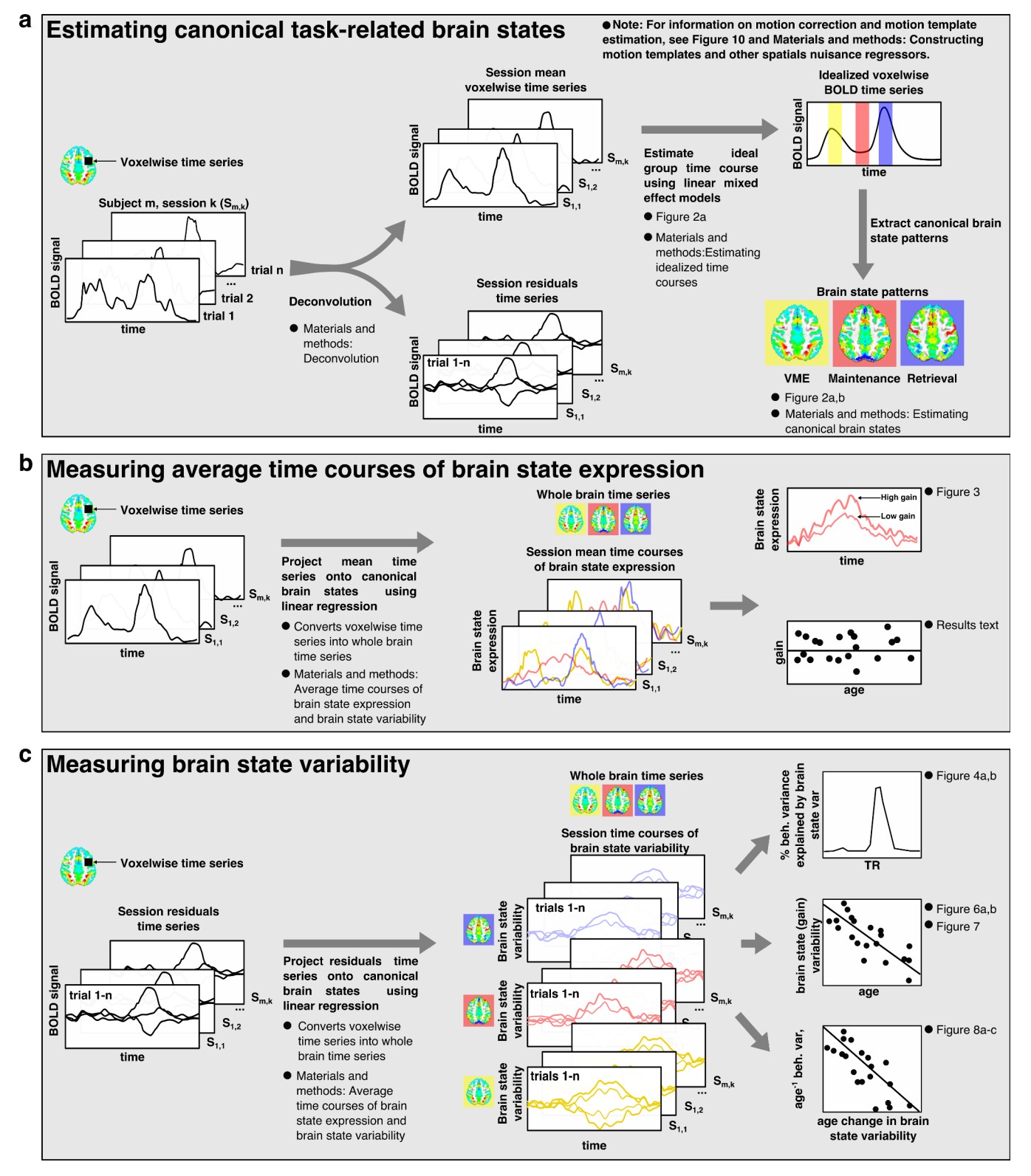

**Figure 9.** Brain state analysis flowchart. A flowchart outlining the major steps involved in the brain state analyses. Where appropriate, the descriptions include a cross-reference to relevant figures depicting major analyses based on the output of each process as well as a cross-reference to the relevant section of Materials and methods for further detailed information on how the process or analysis was implemented.
DOI: https://doi.org/10.7554/eLife.25606.011

Head motion was computed along six affine components corresponding to translation in the three cardinal directions and rotations about three orthogonal axes. In addition we computed a time course of total displacement for each session based on the Euclidean norm of the time derivative of the movement time series at each time point. To account for the prolonged effect of autocorrelated movement on the BOLD signal, we included temporally leading ($-1$TR) and lagging ($+1$–2 TRs) copies of each of the seven motion regressors (*Friston et al., 1996*). Each of the seven motion time courses therefore contributed four motion regressors to the deconvolution design matrix. After deconvolution, we scaled the resulting whole-brain average trial time courses at each voxel, normalizing them to the standard deviation of the regression residuals at the same voxel location.

## Estimating the idealized time courses

Idealized voxel-wise trial time courses for the long delay conditions were estimated from the scaled average trial time course estimates. We modeled these separately for each condition and target hemifield using 3dLME (AFNI), a linear mixed-effects framework. Each time point was modeled as a separate categorical fixed effect and we did not include an intercept term in the model. To account for any bias due to the over representation of subjects who participated in more scans, we included subject identity as a random effect component in the regression model. For each trial type we computed the total displacement undergone by each subject's brain during the BOLD signal measurement intervals (trial durations plus 15 TRs) and included it as a fixed-effect component in the regression analysis. We calculated subjects' average age for all of their sessions and, after centering by the global mean, included it as a subject level fixed-effects regressor. We included a mean age by time interaction term to capture age-related differences in the voxel-wise time courses. We included the subjects' age at each session, after subject level mean-centering, as a second age-related random-effects regressor. Within a given voxel, a single whole trial time course may include independent contributions from visually- and memory-guided saccade events. To account for potential differences due to variability in the number of correct saccades, we included the proportion of unclassifiable and incorrect visually- and memory-guided saccades and their interactions with time as fixed-effect components of the model. We produced idealized trial time courses by generating the voxel-wise model estimates for a subject of mean age, mean in-scanner displacement, and perfect trial performance. This process generated four idealized whole-brain time series corresponding to both long-delay conditions in which targets were located in either the left or right visual hemifield. We used these idealized BOLD time series in our construction of the canonical brain states

## Constructing motion templates and other spatial nuisance regressors

Including lagged motion regressors during deconvolution accounts for the prolonged linear effects of in-scanner motion up to 2 TRs after a given movement (*Friston et al., 1996*). To control for any effects of motion that may continue beyond that time and bias measures of brain state variability, we developed a method to account for temporally prolonged linear motion artifacts from fMRI data based on motion template volumes that model the spatial pattern of artifacts associated with the linear effects of motion (*Figure 10a*). We normalized the regression coefficients associated with each lagged motion regressor at each voxel by the temporal standard deviation of the voxel's post-deconvolution residuals and computed their resulting mean spatial patterns across all subjects. We used these whole-brain patterns of normalized regression coefficients to construct motion artifact templates. For each of the 28 templates (7 motion components with 4 temporal lead/lags), we subtracted the spatial mean of all voxel values and scaled the resulting volumes to a common vector magnitude. Then, using principle component decomposition, we found a set of 11 motion templates that captured >90% of the variability in the set, which were then converted back into 3D volumes.

We verified that the set of motion templates captured components of linear motion related BOLD signal variability that remained after deconvolution by computing $SS_{motion}$, the sum of squared error associated with all motion templates across all TRs in the whole-brain residual BOLD time series. We then computed the ratio $SS_{motion}/(SS_{brain} + SS_{error})$ where $SS_{brain}$ is the sum of squared error associated with brain state variability and $SS_{error}$ is unexplained sum of squared error. We refer to this ratio as motion template variance. We found that estimates of mean FD and motion template variance are highly related (t(337)=10.3; p=1.06e-21), indicating that even with rigorous motion controls during deconvolution (see Materials and methods), significant linear motion artifacts remain

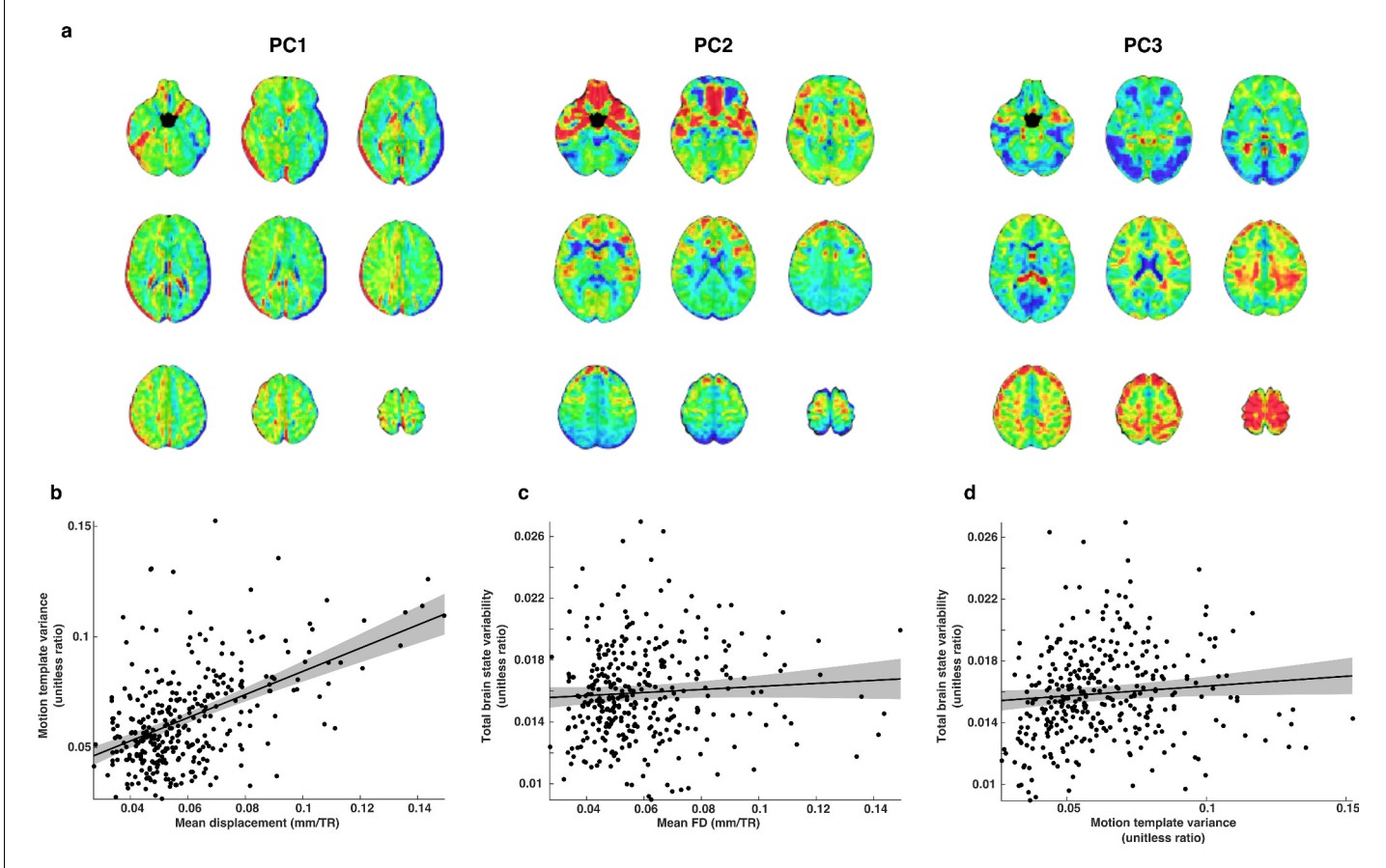

**Figure 10.** The relationship between motion and brain state variability. (a) Illustrative examples of three whole-brain motion templates out of a total set of 19. We used this set of templates as nuisance regressors and to estimate motion related variability that remained in each TR of the whole-brain residual time series. (b) Motion template variance is greater for subjects that moved more in the scanner. (c) Brain state variability was not significantly related to estimates of mean FD. (d) There is a small relationship between motion template variability and brain state variability.
DOI: https://doi.org/10.7554/eLife.25606.012

(*Figure 10b*). We found that while brain state variability is not significantly related to mean FD (t (337)=1.39; p=0.17), it is significantly greater in subjects exhibiting greater motion template variability (t(337)=2.01; p=0.045), (c-d). The usage of motion templates as an additional control for motion-related artifacts arose from an abundance of caution and the need for high dimensional nuisance regressors that could be fit simultaneously to each volume alongside the set of brain state patterns. While these preliminary analyses suggest promise for the approach, we acknowledge that it has yet to be fully validated.

Although the relationship between brain state variability and movement was small and inconsistent across different estimates of motion, we sought a more rigorous control by comparing brain state variability in a group of high motion adults to low motion children and adolescents (see Results).

To remove trivial components of whole-brain BOLD signal variability associated with TR-to-TR shifts in the global mean across the entire imaged volume, we included a constant offset template, which consisted of a whole-brain binary mask. To account for TR-to-TR shifts in mean BOLD signal that are limited to the gray matter (the sort of variability which in some resting state studies is removed via global gray matter signal regression) we included a second binary template defined by the brain state mask (see Materials and methods Estimating canonical brain states).

In addition, we constructed a set of spatial gradient templates to account for other trivial modes of whole-brain BOLD signal variability consisting of simple linear spatial gradient patterns. We first created 3 spatial gradient volumes whose voxel values were equal to their x, y, and z coordinates

relative to the volume's center of mass. We set each voxel that fell outside of a whole-brain MNI mask to zero. We then computed a set of 3 'interaction' templates that corresponded to each pair-wise product of the spatial gradient templates. As a whole, this set of 19 templates constituted the set of spatial nuisance regressors that we used to capture and remove remaining unwanted spatial modes of whole-brain BOLD signal variability associated with motion and trivial variability in global signal.

We included the motion templates and spatial constant and gradient templates as additional nuisance regressors in all analyses that involved projecting whole brain volumes of BOLD signal onto the canonical set of brain state components.

## Estimating canonical brain states

We derived the canonical brain states from whole-brain patterns of BOLD activity extracted from different time points within the idealized trial time courses. To constrain the brain state patterns to swathes of gray matter that were reliably imaged across sessions, we constructed a mask defined by the intersection of a probabilistic MNI gray-matter mask (threshold$\geq$0.5) and a mask constructed from the group average of the model R-squared maps from each session's initial deconvolution (threshold$\geq$0.27, selected to removed unreliably imaged brain regions). Combined these masks served to eliminate white matter; CSF; the hind-brain; cerebellum, the inferior portion of which was not consistently imaged across sessions; artifact-prone basal forebrain; and infero-temporal lobes). The extent of the resulting brain state mask can be observed in (*Figure 2b*).

Each canonical brain state consists of a mean and spatial component. The mean component represents the average pattern of whole-brain BOLD activity evoked by a specific task epoch, regardless of target location. The spatial component represents the difference between the patterns of whole-brain BOLD activity evoked by specific task epochs for right side versus left side targets. To reduce the extent to which BOLD signal resulting from a trial's initial visually-guided saccade contaminated our estimate of maintenance- and retrieval-related brain states, we maximized the temporal distance between the task epochs by using time courses from long delay trials only.

The canonical mean and spatial VME brain states were derived from the whole-brain pattern of activity occurring within the brain state mask around the time of the visually guided saccade. To account for hemodynamic lag, we extracted four volumes of BOLD activity (corresponding to the left and right side targets from both long delay conditions) from the idealized long delay time courses six seconds (4 TRs) after the visually guided saccade was performed to ensure that most regions would have achieved their peak BOLD response (*Handwerker et al., 2004*). We constructed the mean VME brain state by computing the voxel-wise average of normalized BOLD activity across all four volumes. To construct the spatial VME state we separately computed the voxel-wise average of the VME volumes for right and left side targets. The spatial state was defined as the difference –right minus left– of the resulting volumes.

Canonical maintenance-related brain states were based on the patterns of normalized BOLD signal that occurred during the TR immediately prior to a subjects' execution of the memory-guided saccade. Since brain states were estimated from the long delay trial types only, the maintenance brain state patterns were naturally separated from the BOLD response evoked by the initial stimulus presentation and the subjects' subsequent visually guided saccade. To completely remove any remaining stimulus and visuomotor contributions, as well as global average signal, we extracted all volumes from the idealized whole-brain trial time courses that occurred before or included a trial's initial visually guided saccade and regressed these patterns from the maintenance state patterns. States were then constructed for maintenance intervals as described for the VME states above.

Construction of the retrieval-related mean and spatial brain states followed a similar course as that used in the construction of the VME and maintenance-related brain states. Like the VME states, retrieval-related states were based on the normalized BOLD responses occurring during the 4th TR after the execution of the memory-guided saccade. We removed all components of VME and maintenance activity by regressing out every pattern of activity within the brain state mask that occurred before the TR at which the subject began a memory-guided saccade.

The resulting mean and spatial brain states, as expected, exhibited a high degree of mirror symmetry across the midline plane. However, we observed noise and artifacts were introduced into the brain state patterns by accumulating error in our regression-based orthogonalization procedure. To correct for these artifacts, we leveraged symmetry of the brain state patterns by averaging them

with corresponding brain state patterns derived from a left/right mirrored version of the idealized BOLD time series. To do this, we constructed left-right mirrored versions of the idealized trial time courses, performed an identical set of operations to define each brain state and combined them with the original set of brain state patterns. We first applied a mirror matrix to the idealized trial time course volumes. We aligned the mirrored volumes to the standard MNI template using a 12-degrees of freedom affine transformation. We then applied a non-linear transformation that warped the mirrored and aligned volumes to the standard MNI template. The final mean brain state components were constructed by computing the voxel-wise average of the mirrored and non-mirrored mean brain states. We inverted the sign of all voxel values of the mirrored spatial volumes before averaging them with the non-mirrored spatial volumes to produce the final spatial brain state components. In practice the mirroring procedure had only a minor effect on our results, visibly removing noise and improving the relationship that we observed between trial-to-trial behavioral performance and brain state expression (*Figure 4*).

## Average time courses of brain state expression and brain state variability

We converted the average whole-brain trial time series and whole-brain residual time series into average time courses of brain state expression and variability respectively. For each TR, we extracted the whole-brain pattern of activity that we then vectorized and modeled using a linear regression. Our design matrix consisted of vectorized versions of the six brain states (the mean and spatial components of the VME, maintenance and retrieval states) as well as the 19 nuisance regressors templates described above. For each TR we extracted the regression weights for the six brain states, motion, and nuisance components and ordered them into a time series. When this procedure is performed on the whole-brain **average trial time series**, the result is a time course of expression of each of the brain states during a trial. When performed on the whole-brain **residual time series**, the result is a time course of brain state fluctuations, where positive values indicate that a particular brain state was present to a greater extent than average and negative values indicate that a state was expressed less than average. For each session, temporally z-scored the time series of variability for each brain state component.

## Trial-to-trial brain state and behavior relationship

For each session we separately transformed reaction time and saccadic error from each of the four main task conditions into z-scores. SE was rectified such that high SE values reflect greater error in memory-guided saccadic endpoints on a trial. We excluded all trials for which measurements of reaction time and endpoints for both visually- and memory-guided saccades were unavailable due to blink artifacts, noisy data, or transient loss of pupil- or corneal reflection-lock.

We related trial-to-trial variability in reaction time and accuracy to variability in the expression of each brain state across a range of times (±15 TRs) relative to the TRs that contained the memory-guided saccades from each trial (*Figure 4*). Using all correct trials across all sessions, we extracted our z-scored measurements of brain state fluctuation derived from the whole-brain residual time series. We then constructed a regression model that included terms for the measured values of each brain state at the relative TR. We also included terms for the spatial brain state interaction with target hemifield.

Each model contained terms that varied across trials but did not vary across relative TRs. These included terms for run number (coded as 1–3), target hemifield (coded as −1,1), target location (eccentricity, coded from least to most eccentric as 1–3), and the square of target location term. Because it is possible that RT and SE are correlated on a trial-to-trial basis, a true relationship between brain state variability and RT may result in a trivial relationship between brain state variability and SE, or vice versa. To account for this possibility, in the RT regression models, we included a term for trial-to-trial SE and its square. Similarly, for the SE regression models we included a term for RT and its square. This set of regressor terms served as a null model against which the full brain state model was compared. The trial-wise reaction time and accuracy models were fit using a linear mixed-effects framework (MATLAB) to account for the different numbers repeated measurements for many of the subjects. Subject identity was modeled as a random effect. We used the difference between the ordinary $R^2$ values for full and null models at each relative TR to assess the amount of

unique behavioral variability accounted for by trial-to-trial fluctuations in the expression of different brain states. At each relative TR we compared the null and full modes using simulated maximum-likelihood estimation procedure with 5000 iterations (MATLAB).

## Reaction time simulations

To perform these simulations, we compiled a distribution of reaction times for all correct memory-guided saccades across our subject database. We then simulated 400 trials with reaction times selected to produce a distribution that was identical to the compiled empirical distribution. To simulate the simple timing-based effect of BOLD signal variability we generated an impulse function for each simulated trial, a vector where all but one element is equal to zero, where each consecutive element refers to 60 ms time bin after the extinction of the central fixation cross (the signal to perform a memory-guided saccade). For each draw from the reaction time distribution we generated new impulse function vector by inserting a 1 into the vector at the index corresponding to the reaction time on that trial. We convolved each of the 400 impulse functions with a canonical HRF modeled at the same 60 ms resolution. This produced a set of HRF time series whose time of peak amplitude varied with reaction time

Amplitude-based simulations were performed similarly but with two key differences: (1) for each trial we inserted 1 into all impulse function vectors at the same time index, corresponding to mean reaction time, for all trials; and (2) we added or subtracted from the 1 a linearly interpolated value between $\pm 0.25$ where $+0.25$ corresponded to the fastest reaction time and $-0.25$ corresponded to the slowest reaction time. Mixed amplitude and timing based simulations were a hybrid of the two described above. The index of the 1 for each trial's impulse function was selected to coincide with the reaction time on that trial. An additional amplitude modulation factor, as above, was added to the impulse index.

Separately for timing-, amplitude-, and timing and amplitude-based simulations, we computed the mean HRF time series across all trials and simulated residual time series by subtracting the mean HRF time series from the individual trial time series. Next we divided the simulated residuals in to fast and slow RT sets, defined by median split and calculated their average. Lastly, we computed the time integral of the mean residual time series for fast and slow trials.

To compare the simulated pattern of high temporal resolution residuals to the actual data we extracted equivalent snippets (beginning at the TR containing the MGS) of the mean VME brain state fluctuation time course for all trials and all subjects. We selected the mean VME state because of its close relationship with the visuomotor processes which makes it most likely to reflect a trivial timing based relationship between RT and brain state expression. As in the simulated data, for each subject we divided the snippets in to fast and slow RT trials based on a median split and then calculated the group average time series. Finally, we interpolated the resulting time series to a matched temporal resolution using shape preserving piece-wise cubic interpolation (MATLAB).

## Additional information

### Funding

| Funder | Grant reference number | Author |
| --- | --- | --- |
| National Institutes of Health | 5R01MH067924 | David Florentino Montez<br>Finnegan J Calabro<br>Beatriz Luna |
| Staunton Farm Foundation | | Finnegan J Calabro<br>Beatriz Luna |

The funders had no role in study design, data collection and interpretation, or the decision to submit the work for publication.

### Author contributions

David Florentino Montez, Conceptualization, Data curation, Software, Formal analysis, Validation, Investigation, Visualization, Methodology, Writing—original draft, Writing—review and editing; Finnegan J Calabro, Validation, Investigation, Methodology, Writing—review and editing; Beatriz Luna,

Conceptualization, Resources, Supervision, Funding acquisition, Investigation, Methodology, Project administration, Writing—review and editing

### Author ORCIDs
David Florentino Montez (iD) http://orcid.org/0000-0003-1672-8218

### Ethics
Human subjects: Participants and/or their legal guardians provided informed consent before participating in this study. Experimental procedures for this study complied with the Code of Ethics of the World Medical Association (1964; Declaration of Helsinki) and the Institutional Review Board at the University of Pittsburgh. Subjects were paid for their participation in the study.

### Decision letter and Author response
Decision letter https://doi.org/10.7554/eLife.25606.016
Author response https://doi.org/10.7554/eLife.25606.017

## Additional files

### Supplementary files
• Transparent reporting form
DOI: https://doi.org/10.7554/eLife.25606.013

### Major datasets
The following dataset was generated:

| Author(s) | Year | Dataset title | Dataset URL | Database, license, and accessibility information |
|---|---|---|---|---|
| David Florentino Montez, Finnegan J Calabro, Beatriz Luna | 2017 | Data from: The expression of established cognitive brain states stabilizes with working memory development | http://dx.doi.org/10.5061/dryad.68ff1 | Available at Dryad Digital Repository under a CC0 Public Domain Dedication |

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
