## [Decision Letter]

Thank you for submitting your article "Stable engagement of cognitive brain states underlies the maturation of working memory" for consideration by *eLife*. Your article has been reviewed by two peer reviewers, and the evaluation has been overseen by Sabine Kastner as the Reviewing Editor and Senior Editor. The following individual involved in review of your submission has agreed to reveal her identity: Silvia Bunge (Reviewer #1).

The reviewers have discussed the reviews with one another and the Reviewing Editor has drafted this decision to help you prepare a revised submission.

Summary and Essential revisions:

The reviewers and editor thought that this is an interesting and methodologically rigorous paper examining the neural basis of reduced variability in cognitive performance over the course of development. The authors measure brain state variability while participants across a wide age range perform a memory-guided saccade task, and show that this variability declines with age. They attempt to parse the task trials into different components and argue that it is particularly the delay-period activation that is showing an age-related decrease in brain state variability over time.

The reviewers were enthusiastic about the paper. However, they thought that the methods were often hard to understand, and it would be difficult to replicate the study with its present methods description. Therefore, we are asking you to thoroughly revise particularly the Methods section and improve its clarity. There are many details missing, the analysis pipeline needs better justification. Please find the reviews appended for further suggestions.

Reviewer #1:

This is an interesting and methodologically rigorous paper examining the neural basis of reduced variability in cognitive performance over the course of development. The authors measure brain state variability while participants across a wide age range perform a memory-guided saccade task, and show that this variability declines with age. They attempt to parse the task trials into different components (the initial encoding, the delay period – and more specifically the delay-period activity that represents the spatial information – and the memory-guided saccade) and argue that it is particularly the delay-period activation that is showing an age-related decrease in brain state variability over time. Critically, the authors show that their results cannot be accounted for by reduced head motion. They hypothesize that these results reflect increased neural gain over the course of development.

The Introduction and Discussion sections are very clearly written. I did not follow the methods section all the way through, although it is entirely possible that someone with more familiarity with these methods would find it clearly written. I found the figure legends insufficiently detailed, and wasn't always sure what I was looking at. I have pointed to a few parts of the manuscript that would warrant clarification, but would recommend going through it carefully one more time to identify gaps.

I am always a little wary of efforts to separate the encoding and maintenance periods of a working memory task, as it seems artificial – and orthogonalizing these components seems all the more so. However, I understand why the authors have done this, and have no specific concerns.

Overall, I think that this paper will make a valuable contribution to our understanding of cognitive development.

Reviewer #2:

This report is from a strong group with a very impressive dataset that sets to show that adolescent working memory is supported by specified brain states during these tasks. Specifically, brain state variability is tightly linked to the variance and development of task performance. The work is certainly novel and the topic is important; however, it is extremely dense and very difficult to follow. The logic and approach here are not very straightforward, and very hard to understand, which would make a future replication of the findings quite difficult. At the same time it's also not clear that the construct in question is actually being directly examined by the methods. I will admit that I am only one reviewer that potentially is simply missing something, because the general idea and findings are interesting, I just found it very difficult to evaluate.

More effort needs to be done in describing why what is being done here represents "brain states."

Is there any analysis validating the procedures to decompose the patterns of brain activity. In other words, how do we know what is being described works with regard to isolating the signals? For example, peak activity in the BOLD responses is assumed to occur after 6 seconds, but we know this is highly variable for tasks and brain regions. It's just hard to know whether what is being done to do this is actually disentangling these processes. There are simply a lot of assumptions throughout the manuscript, and what seem to be arbitrary decisions regarding the analysis stream, without any clear rationale or validation.

Along the same lines, the authors did a good job at handling various confounds, such as motion, but again, the details and procedures for exactly how this was being done are not there, nor am I aware that the procedures have been validated elsewhere. As the authors' own analysis shows, getting this right for this type of analysis is quite important.

All figures/plots (including Figure 1) of the model fits need to include the raw data to be able to visualize the quality of the fits.

Also need to include scales on all of the images. They should also be consistent within an image.

Overall, I thought the work was interesting, but simply hard to follow and difficult to evaluate rigorously.

[Editors' note: further revisions were requested prior to acceptance, as described below.]

Thank you for resubmitting your work entitled "The expression of established cognitive brain states stabilizes with working memory development" for further consideration at *eLife*. Your revised article has been favorably evaluated by Sabine Kastner (Senior & reviewing editor), and two reviewers (Silvia Bunge & Damien Fair).

The manuscript has been improved but there are some remaining issues that need to be addressed before acceptance, as outlined below:

Reviewer #2:

The authors did a relatively strong job at making things more clear. It is still a very dense manuscript, but again the concepts are strong and the report is important. It should be accepted and will be a very nice contribution.

As the authors know, this reviewer is very concerned about motion having an effect on these types of analyses for several reasons, but based on the recent literature (e.g. Siegel et al., 2014), the findings here would be expected based on how movement affects task related signals. While the authors provide some more citations related to their motion correction procedures, they are often older citations that come prior to many of the recent reports on the issues with using traditional translation and rotation numbers to quantify the impact of motion on the BOLD signals. I won't belabor that here. In addition, the spatial template procedure here, while novel doesn't account for the non-spatial artifacts of motion (see Siegel et al., 2016; Burgess et al., 2016, amongst others). I do not want this to take away from my overall enthusiasm for the work here. It is quite strong and very interesting, I'm just overly cautious. What would be great for the reviewers to do is two-fold. First, simply point out anywhere that the while you were overly cautious on controlling for motion here, future work will have to done validate the efficacy of the procedures. Second, the analysis for Figure 7 (which I thought was the most convincing actually) should be re-done, but rather than matching on the traditional translation and rotation numbers, the matching should be done using a mean Frame-to-frame displacement (As in Siegel, 2014; Power, 2012). I only say this because matching based on traditional translation and rotation numbers doesn't always provide groups as you might expect with regard to motion (this is the problem with many of the papers in the connectivity literature, they are "matching" on the wrong parameters). Again, just being overly cautious here. Perhaps this suggestion was already what was used for making high and low motion groups; however, what measures were actually used for this analysis I can't tell based on the results or methods sections. If it was done this way already, then, just clarify.

---

## [Author Response]

*Reviewer #1:*

*This is an interesting and methodologically rigorous paper examining the neural basis of reduced variability in cognitive performance over the course of development. The authors measure brain state variability while participants across a wide age range perform a memory-guided saccade task, and show that this variability declines with age. They attempt to parse the task trials into different components (the initial encoding, the delay period – and more specifically the delay-period activity that represents the spatial information – and the memory-guided saccade) and argue that it is particularly the delay-period activation that is showing an age-related decrease in brain state variability over time. Critically, the authors show that their results cannot be accounted for by reduced head motion. They hypothesize that these results reflect increased neural gain over the course of development.*

We thank the reviewer for indicating that this important aspect of our study was not clear. We have now relabeled the relevant section of the results to clarify that, critically, we did not observe evidence for a change in *mean* neural gain over the course of development; rather we observed a reduction in neural gain *variability.*

*The Introduction and Discussion sections are very clearly written. I did not follow the methods section all the way through, although it is entirely possible that someone with more familiarity with these methods would find it clearly written. I found the figure legends insufficiently detailed, and wasn't always sure what I was looking at. I have pointed to a few parts of the manuscript that would warrant clarification, but would recommend going through it carefully one more time to identify gaps.*

We have now edited the text to provide greater clarity and added a flow chart (Figure 9) for inclusion in the methods section that provides a broad outline of the concept, process, and goals of the brain state analyses. We hope this figure will provide a convenient simplified framework for interpreting the detailed methods sections and facilitate replication.

*I am always a little wary of efforts to separate the encoding and maintenance periods of a working memory task, as it seems artificial – and orthogonalizing these components seems all the more so. However, I understand why the authors have done this, and have no specific concerns.*

We agree with Reviewer 1 and acknowledge that the separation of encoding and maintenance periods is somewhat artificial. We now clarify the importance of identifying the brain state patterns associated with visuomotor processes, i.e. visually-guided eye movements that critically distinguish the encoding period from maintenance processes. In addition, in the Discussion section we acknowledge that our task design would not, in principle, allow us to dissociate putative encoding processes from visuomotor processes and therefore refer to the visuomotor state as visuomotor/encoding (VME).

*Reviewer #2:*

*This report is from a strong group with a very impressive dataset that sets to show that adolescent working memory is supported by specified brain states during these tasks. Specifically, brain state variability is tightly linked to the variance and development of task performance. The work is certainly novel and the topic is important; however, it is extremely dense and very difficult to follow. The logic and approach here are not very straightforward, and very hard to understand, which would make a future replication of the findings quite difficult. At the same time it's also not clear that the construct in question is actually being directly examined by the methods. I will admit that I am only one reviewer that potentially is simply missing something, because the general idea and findings are interesting, I just found it very difficult to evaluate.*

We agree that the paper is very dense; we have incurred a significant explanatory burden in attempting to report novel scientific finding as well as the new methods by which they were revealed. Significant changes have been made throughout the body of the manuscript and the methods section to clarify our approach and rationale. We have also included an addition explanatory figure (Figure 9), which outlines the major goals and analyses that we performed and provides a conceptual flowchart that links the major steps in the analyses to the relevant results figures. We hope that the inclusion of this figure will provide a better conceptual framework by which readers can interpret the individual sections of the Materials and methods.

*More effort needs to be done in describing why what is being done here represents "brain states."*

We have reworked the text to better clarify why our approach defines a brain state and what we mean by using this term. In addition, edits to Figure 2 now elaborate on the method of estimating brain state patterns and depict graphically what is represented by a brain state by juxtaposing the time courses of individual exemplar voxels that intuitively exhibit activity that associating them with a particular component of the task alongside the brain states derived from the idealized BOLD time series.

*Is there any analysis validating the procedures to decompose the patterns of brain activity. In other words, how do we know what is being described works with regard to isolating the signals? For example, peak activity in the BOLD responses is assumed to occur after 6 seconds, but we know this is highly variable for tasks and brain regions. It's just hard to know whether what is being done to do this is actually disentangling these processes. There are simply a lot of assumptions throughout the manuscript, and what seem to be arbitrary decisions regarding the analysis stream, without any clear rationale or validation.*

The time courses depicted in Figure 3 provide validation that our procedure isolates signals associated with visuomotor, maintenance, and retrieval. For instance, examination of the time course of the VME states in the long delay trials reveals two peaks of expression, associated with the VGS and MGS events; the maintenance state is expressed maximally between the occurrence of the VGS and MGS; and lastly, the retrieval state is expressed only during the MGS, indicating that we have successfully removed the components associated with the visuomotor processes which occur during the VGS. Importantly, all subjects regardless of age similarly express the canonical brain states, on average.

*Along the same lines, the authors did a good job at handling various confounds, such as motion, but again, the details and procedures for exactly how this was being done are not there, nor am I aware that the procedures have been validated elsewhere. As the authors' own analysis shows, getting this right for this type of analysis is quite important.*

We dealt with potential motion confounds in several ways, many of which have been employed in other studies. We also develop an additional method of estimating the magnitude of residual motion artifacts that is idiosyncratic to our brain state approach. We thank this reviewer for pointing out the need for additional citations for those approaches that have historical precedent as well as pointing out the need for additional clarification on and validation of the approach that we developed.

First, we exclude sessions in which average displacement per TR exceeded 2.0 mm. Second, we used leading and lagging motion regressors to estimate and account for the prolonged and autocorrelated effects of motion as outlined previously in (Friston et al., 1996). Importantly, it is algebraically equivalent to including higher order derivatives of the motion time series as nuisance regressors, which is a common practice in fMRI time series analysis. Lastly, as noted in Materials and methods, the regression models that estimate the relationship between age and measures of brain state variability include summary measure of in-scanner motion as a nuisance covariate.

We verify brain state variability is greater in a group of younger, low motion subjects and a group of older, high motion subjects. The algorithm that we employ to generate these biased distributions is a minor variation on the mean matching procedures (based on an intersection of histograms approach) that are often used to match firing rates as a control in the electrophysiological experiments, e.g. (Churchland et al., 2010; Churchland et al., 2007; Cohen & Maunsell, 2009). We have now included these references.

Our novel contribution to controlling for motion confounds is the construction of motion templates derived from the estimated motion time series. These templates are primarily useful when fitting individual volumes (TRs) to the set of brain state patterns as they capture residual whole-brain BOLD signal variance known to be associated with movement. Their inclusion slightly improved the measured relationship between trial-to-trial brain state expression and behavior (Figure 4).

While we do think that this approach, or a variation on it, may provide some conceptual advantage to PCA- or ICA-based methods (particularly because the motion templates are empirically related to movement and require no interpretation), in this study however, we use these simply as additional nuisance regressors in the trial-to-trial brain state/behavior analyses and as a secondary measure of motion-related BOLD signal artifacts to provide an alternative motion-related nuisance regressor. That this method is effective for removing linear effects of motion remaining after deconvolution is demonstrated in Figure 10. This figure shows that the proportion of whole-brain variability associated with the motion templates is highly correlated motion estimates.

*All figures/plots (including Figure 1) of the model fits need to include the raw data to be able to visualize the quality of the fits.*

We have now included the individual data points on these figures.

*Also need to include scales on all of the images. They should also be consistent within an image.*

We have now included a color bar scale on the figures depicting the canonical task-related brain state patterns and note that they are normalized to a common magnitude.

[Editors' note: further revisions were requested prior to acceptance, as described below.]

*Reviewer #2:*

*As the authors know, this reviewer is very concerned about motion having an effect on these types of analyses for several reasons, but based on the recent literature (e.g. Siegel et al., 2014), the findings here would be expected based on how movement affects task related signals. While the authors provide some more citations related to their motion correction procedures, they are often older citations that come prior to many of the recent reports on the issues with using traditional translation and rotation numbers to quantify the impact of motion on the BOLD signals. I won't belabor that here. In addition, the spatial template procedure here, while novel doesn't account for the non-spatial artifacts of motion (see Siegel et al., 2016; Burgess et al., 2016, amongst others). I do not want this to take away from my overall enthusiasm for the work here. It is quite strong and very interesting, I'm just overly cautious. What would be great for the reviewers to do is two-fold. First, simply point out anywhere that the while you were overly cautious on controlling for motion here, future work will have to done validate the efficacy of the procedures. Second, the analysis for Figure 7 (which I thought was the most convincing actually) should be re-done, but rather than matching on the traditional translation and rotation numbers, the matching should be done using a mean Frame-to-frame displacement (As in Siegel, 2014; Power, 2012). I only say this because matching based on traditional translation and rotation numbers doesn't always provide groups as you might expect with regard to motion (this is the problem with many of the papers in the connectivity literature, they are "matching" on the wrong parameters). Again, just being overly cautious here. Perhaps this suggestion was already what was used for making high and low motion groups; however, what measures were actually used for this analysis I can't tell based on the results or methods sections. If it was done this way already, then, just clarify.*

To address the first point, we have now included the following passage in the section that discusses the motion template method:

“The usage of motion templates as an additional control for motion-related artifacts arose from an abundance of caution and the need for high dimensional nuisance regressors that could be fit simultaneously to each volume alongside the set of brain state patterns. While these preliminary analyses suggest promise for the approach, we acknowledge that it has yet to be fully validated. (subsection "2.3 Constructing motion templates and other spatial nuisance regressors”).

In regards to the second suggestion, Reviewer #2’s point about the value of using a traditional estimate of in scanner motion as a nuisance covariate is well taken and we recognize that our use of an idiosyncratic measure of motion would have served as an unnecessary distraction for readers of an already complex paper and made it more difficult to contrast with the extant literature. We now use mean FD to quantify in-scanner motion for all of our analyses (as calculated in Power, 2012). In practice, mean FD is highly correlated (essentially differing by scaling factor) with the values that we had used to quantify motion in our initial analyses (r=0.9977; p~0; across all sessions). Thus, our results differ only nominally. We selected a conservative motion threshold of mean FD <= 0.15mm which allowed us to keep a nearly identical same data set. As a bonus, applying this selection criterion resulted in the inclusion of three additional sessions of data that were not in the original analyses. The effect on our results were minimal and are reflected in the slight differences in reported statistics and their corresponding degrees of freedom throughout the paper. In order to have a consistent metric for motion throughout the paper, all relevant figures have been regenerated from the resulting data set and group-level analyses now use mean FD as a motion-related nuisance regressor. In addition we include source citations for the calculation of mean FD.